# Sparse Multiple Kernel Learning: Alternating Best Response and Semidefinite Relaxations

**Dimitris Bertsimas**  *dbertsim@mit.edu*
*Massachusetts Institute of Technology*
*Cambridge, MA 02139, USA*

**Caio de Próspero Iglesias**  *caiopigl@mit.edu*
*Massachusetts Institute of Technology*
*Cambridge, MA 02139, USA*

**Nicholas A. G. Johnson**  *nagj@mit.edu*
*Massachusetts Institute of Technology*
*Cambridge, MA 02139, USA*

**Reviewed on OpenReview:** *https://openreview.net/forum?id=Y5icwFwkyh*

## Abstract

We study Sparse Multiple Kernel Learning (SMKL), which is the problem of selecting a *sparse* convex combination of prespecified kernels for support vector binary classification. Unlike prevailing $\ell_1$-regularized approaches that approximate a sparsifying penalty, we formulate the problem by imposing an explicit cardinality constraint on the kernel weights and add an $\ell_2$ penalty for robustness. We solve the resulting non-convex minimax problem via an *alternating best response* algorithm with two subproblems: the $\boldsymbol{\alpha}$-subproblem is a standard kernel SVM dual solved via LIBSVM, while the $\boldsymbol{\beta}$-subproblem admits an efficient solution via the Greedy Selector and Simplex Projector algorithm. We reformulate SMKL as a mixed integer semidefinite optimization problem and derive a hierarchy of semidefinite convex relaxations which can be used to certify near-optimality of the solutions returned by our best response algorithm and also to warm start it. On ten UCI benchmarks, our method with random initialization outperforms state-of-the-art MKL approaches in out-of-sample prediction accuracy on average by 3.34 percentage points (relative to the best performing benchmark) while selecting a small number of candidate kernels in comparable runtime. With warm starting, our method outperforms the best performing benchmark's out-of-sample prediction accuracy on average by 4.05 percentage points. Our convex relaxations provide a certificate that in several cases, the solution returned by our best response algorithm is the globally optimal solution.

## 1 Introduction

Given a dataset $\{(\boldsymbol{x}_i, y_i)\}_{i=1}^n$ where $\boldsymbol{x}_i \in \mathbb{R}^m$ are $m$-dimensional features and $y_i \in \{-1, 1\}$ are binary labels, the Kernel Support Vector Machine problem (KSVM) seeks to select a possibly infinite-dimensional feature map $\Phi : \mathbb{R}^m \to \mathbb{R}^d$ and learn a linear classification rule $\hat{y} = \mathrm{sgn}(\boldsymbol{w}^T \Phi(\boldsymbol{x}) + b)$ that generalizes well to unseen data. For a fixed feature map $\Phi$, the learning problem can be written in primal form as:

$$\min_{\boldsymbol{w} \in \mathbb{R}^d, \boldsymbol{\eta} \in \mathbb{R}_+^n, b \in \mathbb{R}} \quad \frac{1}{2} \|\boldsymbol{w}\|_2^2 + C \sum_{i=1}^n \eta_i, \quad \text{s.t.} \quad y_i(\boldsymbol{w}^T \Phi(\boldsymbol{x}_i) + b) \geq 1 - \eta_i \quad \forall i, \tag{1}$$

where $C \in \mathbb{R}_+$ is a hyperparameter that is to be cross-validated by minimizing a validation metric (see, e.g., Owen & Perry, 2009) to obtain strong out-of-sample performance in theory and practice (Bousquet &

Elisseeff, 2002). The first term in the objective function of equation 1 encourages minimal norm solutions as a regularity condition that reduces overfitting while the second term in the objective function penalizes misclassified examples in the training dataset.

The KSVM learning problem given by equation 1 emits a dual formulation that is often the preferred formulation to consider as it avoids requiring explicit construction of the feature map $\Phi$ (which may be prohibitive for large dimensional feature maps or simply impossible in the case of infinite dimensional feature maps). Specifically, the dual form of KSVM is given by

$$\max_{\boldsymbol{\alpha} \in \mathbb{R}_+^n} \quad \sum_{i=1}^n \alpha_i - \frac{1}{2}(\boldsymbol{y} \circ \boldsymbol{\alpha})^T \boldsymbol{K}(\boldsymbol{y} \circ \boldsymbol{\alpha}), \quad \text{s.t.} \quad \sum_{i=1}^n \alpha_i y_i = 0, \quad C \geq \alpha_i \geq 0 \quad \forall i, \tag{2}$$

where $\boldsymbol{K}$ is the kernel matrix corresponding to the feature map $\Phi$ (meaning that $K_{ij} = \Phi(\boldsymbol{x}_i)^T \Phi(\boldsymbol{x}_j)$).

In vanilla KSVM, the choice of kernel matrix is typically selected via cross-validation. This however is inefficient and significantly limits the number of kernels that are considered. Moreover, it can be natural to use different kernels for different features. This might occur, for instance, if each data point consists of both text-based and visual features. Multiple kernel learning (MKL) addresses this limitation by considering a prespecified collection of $q$ candidate kernels $\{\boldsymbol{K}_i\}_{i=1}^q$ and searching over the space of convex combinations of said kernels (note that some treatments of MKL consider general conic combinations of candidate kernels rather than only convex combinations). Explicitly, MKL is formulated as:

$$\min_{\boldsymbol{\beta} \in \mathbb{R}_+^q} \max_{\boldsymbol{\alpha} \in \mathbb{R}_+^n} \quad \sum_{i=1}^n \alpha_i - \frac{1}{2}(\boldsymbol{y} \circ \boldsymbol{\alpha})^T \left[ \sum_{i=1}^q \beta_i \boldsymbol{K}_i \right] (\boldsymbol{y} \circ \boldsymbol{\alpha})$$
$$\text{s.t.} \quad \sum_{i=1}^n \alpha_i y_i = 0, \quad C \geq \alpha_i \geq 0 \quad \forall i, \quad \|\boldsymbol{\beta}\|_1 = 1. \tag{3}$$

Since any symmetric positive semidefinite matrix is a valid kernel (meaning that it corresponds to some possibly infinite dimensional feature map), the matrix $\boldsymbol{K}(\boldsymbol{\beta}) = \sum_{i=1}^q \beta_i \boldsymbol{K}_i$ is a valid kernel matrix for any $\boldsymbol{\beta} \in \mathbb{R}_+^q$. In words, equation 3 seeks to find the kernel in the convex hull of the $q$ kernels $\{\boldsymbol{K}_i\}_{i=1}^q$ that results in the lowest training objective value when used to fit KSVM. The inclusion of the constraint $\|\boldsymbol{\beta}\|_1 = 1$ in equation 3 is typically motivated as inducing sparsity in $\boldsymbol{\beta}$ as a convex surrogate to the $\ell_0$ norm function. In this work, we take the approach of directly imposing a cardinality constraint on $\boldsymbol{\beta}$ and demonstrate that this yields a tractable formulation that produces superior solutions to MKL. Explicitly, given a target sparsity level $k_0 \in \mathbb{N}$, we consider:

$$\min_{\boldsymbol{\beta} \in \mathbb{R}_+^q} \max_{\boldsymbol{\alpha} \in \mathbb{R}_+^n} \quad \sum_{i=1}^n \alpha_i - \frac{1}{2}(\boldsymbol{y} \circ \boldsymbol{\alpha})^T \left[ \sum_{i=1}^q \beta_i \boldsymbol{K}_i \right] (\boldsymbol{y} \circ \boldsymbol{\alpha}) + \lambda \|\boldsymbol{\beta}\|_2^2$$
$$\text{s.t.} \quad \sum_{i=1}^n \alpha_i y_i = 0, \quad C \geq \alpha_i \geq 0 \quad \forall i, \quad \|\boldsymbol{\beta}\|_1 = 1, \quad \|\boldsymbol{\beta}\|_0 \leq k_0, \tag{4}$$

where in addition to the cardinality constraint on $\boldsymbol{\beta}$ we have introduced an $\ell_2$ regularization term in the objective function to encourage robustness.

## 1.1 Contribution and Structure

In this paper, we present an alternating best response algorithm that efficiently produces solutions to equation 4 which outperform state-of-the-art benchmark methods. We exactly reformulate equation 4 as a mixed integer semidefinite program which emits a natural hierarchy of semidefinite convex relaxations that produce lower bounds on the optimal objective value of equation 4. We employ these lower bounds to produce certificates of near optimality of the feasible solutions returned by our alternating best response algorithm and additionally as warm starts for our algorithmic approach. We present rigorous numerical results across

UCI benchmark datasets. On average, our approach outperforms the best performing benchmark method in out-of-sample prediction accuracy by 3.34 percentage points while selecting a small number of candidate kernels in comparable runtime when using a random initialization. With warm starting, our approach outperforms the best performing benchmark's out-of-sample prediction accuracy by 4.05 percentage points on average. In many cases, the lower bounds produced by our semidefinite relaxations certify that the solution returned by our alternating best response heuristic is the globally optimal solution to equation 4.

The rest of the paper is laid out as follows. In Section 2, we review previous work that is closely related to SMKL. In Section 3, we study the two subproblems that emerge naturally from equation 4 and we present our alternating best response algorithm. We reformulate equation 4 as a mixed integer semidefinite optimization problem in Section 4 and present a family of convex relaxations. Finally, in Section 5 we investigate the performance of our algorithm against benchmark methods on real world UCI datasets.

**Notation:** We let nonbold face characters such as $b$ denote scalars, lowercase bold faced characters such as $\boldsymbol{x}$ denote vectors, uppercase bold faced characters such as $\boldsymbol{X}$ denote matrices, and calligraphic uppercase characters such as $\mathcal{Z}$ denote sets. We let $[n]$ denote the set of running indices $\{1, ..., n\}$. We let $\boldsymbol{0}_n$ denote an $n$-dimensional vector of all 0's, $\boldsymbol{0}_{n \times m}$ denote an $n \times m$-dimensional matrix of all 0's, and $\boldsymbol{I}_n$ denote the $n \times n$ identity matrix. We let $\mathcal{S}^n$ denote the cone of $n \times n$ symmetric matrices and $\mathcal{S}^n_+$ denote the cone of $n \times n$ positive semidefinite matrices.

## 2 Literature Review

In this section, we present a non-exhaustive literature review to highlight key ideas from the MKL literature that we build upon with a particular focus on open source methods that are used as benchmarks in this work. As an exhaustive review of the literature is outside the scope of this paper, we refer the interested reader to Gönen & Alpaydın (2011). In Appendix A, we review commonly used kernel families that we leverage in this work.

### 2.1 MKL Foundations

Multiple Kernel Learning emerged in the early 2000s as a framework to learn an optimal combination of prespecified base kernels rather than relying on a single, preselected kernel. Lanckriet et al. (2004) first formulated MKL as a semidefinite optimization problem that learned nonnegative kernel weights alongside an SVM classifier. The authors demonstrated that combining kernels from heterogeneous data sources can significantly improve accuracy over the best single kernel. As off the shelf interior point methods suffer from scalability challenges for semidefinite programming, several subsequent efforts sought to improve the scalability of MKL by leveraging alternate formulations and algorithmic methods. To this end, Bach et al. (2004) reformulated MKL as a quadratically constrained quadratic program and designed a sequential minimization based solution method. In order to solve MKL problems with large datasets, Sonnenburg et al. (2006) developed a column generation algorithm to solve a semi-infinite linear programming reformulation of MKL and Rakotomamonjy et al. (2008) developed the widely used SimpleMKL solver which leverages an efficient gradient descent based algorithm.

Recognizing that the number of base kernels to consider may often be considerably large, several authors sought to develop techniques that produced a sparse kernel meaning that the selected kernel was formed using only a small subset of the input base kernels. As directly imposing a cardinality constraint on the kernel combination vector $\boldsymbol{\beta}$ produces an NP hard problem, many approaches leverage the $\ell_1$ norm function as a convex surrogate for the $\ell_0$ norm function analogous to the approach taken by Lasso to solve sparse regression (Tibshirani, 1996). To this end, Bach (2008) proposed an MKL formulation that sought to enforce group-level sparsity through the inclusion of a weighted $\ell_1$-norm regularizer in the SVM dual, which was shown to have desirable consistency properties. Several other convex approximations have been employed in the literature. For example, Koltchinskii & Yuan (2010) attempts to induce sparsity through a combination of an empirical $\ell_2$ norm and the reproducing kernel Hilbert space norms induced by the base kernels. In an alternate direction, Kloft et al. (2011) explore $\ell_p$ norm constraints (for $p \geq 1$) and theoretically explore their influence on sparsity.

Beyond $\ell_p$ surrogates, some authors have explored non-convex penalties that more closely approximate the $\ell_0$ norm. For example, Subrahmanya & Shin (2010) propose a log-penalised MKL formulation that attains competitive accuracy on cancer-detection and hyperspectral-image benchmarks while selecting markedly fewer kernels than other multiple kernel methods. Nevertheless, the sparsity in their approach is still induced through a smooth relaxation rather than an explicit cardinality constraint.

Despite their empirical success, smooth or convex surrogates of $\ell_0$ do not guarantee truly sparse solutions. Lasso-type penalties, for instance, produce robust estimators but not sparse estimators and can be out-performed by methods that exactly enforce sparsity (Bertsimas & Copenhaver, 2018). This insight has motivated a recent line of work that revisits direct cardinality constraints in MKL. Xue & Song (2019) propose a non-convex formulation of sparse MKL which they solve using techniques from difference-of-convex programming. However, their approach considers using various smooth approximations to the $\ell_0$ norm (exponential, logarithmic, capped $\ell_1$ and SCAD). In this work, we consider an explicit $\ell_0$ norm constraint which we enforce directly without approximation.

## 2.2 Open Source MKL Benchmarks

In this work, we benchmark our methods against three well studied and widely used MKL algorithms that are implemented in the open-source `MKLpy` library (Lauriola & Aiolli, 2020): *AverageMKL*, *EasyMKL* (Aiolli & Donini, 2015), and *CKA* (Cortes et al., 2012). We briefly outline each of these approaches below.

**AverageMKL** This baseline sets all kernel weights uniformly as $\beta_i = \frac{1}{q}$ for all $i$. This produces the following combined kernel:

$$\boldsymbol{K}(\boldsymbol{\beta}) = \frac{1}{q} \sum_{i=1}^{q} \boldsymbol{K}_i.$$

Although AverageMKL trivially determines kernel weights, this method is surprisingly competitive when most kernels are informative (Lewis et al., 2006). Its parameter-free design and limited training cost make it a useful sanity check and benchmark for evaluating the performance of more sophisticated MKL methods.

**EasyMKL** This method is a scalable MKL algorithm that learns a combined kernel without alternating between SVM training and kernel weight updates. Explicitly, EasyMKL seeks a combined kernel $\boldsymbol{K}(\boldsymbol{\beta}) = \sum_{i=1}^{q} \beta_i \boldsymbol{K}_i$ where $\boldsymbol{\beta} \geq 0$ and $\|\boldsymbol{\beta}\|_2^2 = 1$. This is achieved by introducing a distribution vector $\boldsymbol{\gamma} \in \Gamma$ over the training examples (separately normalized on positive and negative classes), and letting $\boldsymbol{Y} = \mathbf{diag}(y)$. Here, $\Gamma$ denotes the set of possible distributions. Define $\boldsymbol{d}(\boldsymbol{\gamma}) \in \mathbb{R}^q$ by letting $d_i(\boldsymbol{\gamma}) = \boldsymbol{\gamma}^\top \boldsymbol{Y} \boldsymbol{K}_i \boldsymbol{Y} \boldsymbol{\gamma}$, the (squared) margin of $\boldsymbol{K}_i$ under weighting $\boldsymbol{\gamma}$. EasyMKL casts MKL as the saddle-point problem

$$\min_{\boldsymbol{\gamma} \in \Gamma} \max_{\|\boldsymbol{\beta}\|_2 = 1} \left[ (1 - \lambda) \, \boldsymbol{\beta}^\top d(\boldsymbol{\gamma}) \, + \, \lambda \, \|\boldsymbol{\gamma}\|_2^2 \right], \tag{5}$$

where $\lambda \in [0, 1]$ trades off margin maximization against variance regularization. For fixed $\boldsymbol{\gamma}$, the inner maximization admits the closed-form solution $\boldsymbol{\beta}^* = \frac{d(\boldsymbol{\gamma})}{\|d(\boldsymbol{\gamma})\|_2}$ which reduces equation 5 to

$$\min_{\boldsymbol{\gamma} \in \Gamma} \left[ (1 - \lambda) \, \|d(\boldsymbol{\gamma})\|_2 \, + \, \lambda \, \|\boldsymbol{\gamma}\|_2^2 \right]. \tag{6}$$

The authors then replace $\|d(\boldsymbol{\gamma})\|_2$ in equation 6 with $\|d(\boldsymbol{\gamma})\|_1$, which produces a more tractable optimization problem and empirically is sparsity inducing.

**CKA (Centered Kernel Alignment)** This approach selects kernel weights by directly maximizing the alignment between a convex combination of base kernels and the target label kernel. Let $\boldsymbol{K}_0 = \boldsymbol{y}\boldsymbol{y}^\top$ and denote by $\boldsymbol{K}_i^c = \boldsymbol{C} \boldsymbol{K}_i \boldsymbol{C}$ the centered version of kernel $\boldsymbol{K}_i$ where $\boldsymbol{C} = \boldsymbol{I} - \frac{1}{n}\mathbf{1}\mathbf{1}^T$. Let $\boldsymbol{a} = \left( \langle \boldsymbol{K}_1^c, \boldsymbol{K}_0^c \rangle, \ldots, \langle \boldsymbol{K}_q^c, \boldsymbol{K}_0^c \rangle \right)^\top$ and let $M_{ij} = \langle \boldsymbol{K}_i^c, \boldsymbol{K}_j^c \rangle$. Then CKA solves the generalized Rayleigh quotient

$$\max_{\boldsymbol{\beta} \geq 0, \, \|\boldsymbol{\beta}\|_2 = 1} \frac{(\boldsymbol{\beta}^\top \boldsymbol{a})^2}{\boldsymbol{\beta}^\top \boldsymbol{M} \boldsymbol{\beta}},$$

whose solution admits the closed-form $\boldsymbol{\beta}^* = \boldsymbol{M}^{-1}\boldsymbol{a}/\|\boldsymbol{M}^{-1}\boldsymbol{a}\|_2$. While CKA typically produces dense weight vectors, its efficiency and strong empirical performance make it a valuable baseline for MKL.

In summary, AverageMKL serves as a simple and effective baseline, EasyMKL provides a sparsity inducing and scalable convex method grounded in margin maximization, and CKA offers a fast alignment-based solution. The strong empirical performance and reproducibility of these methods make them well-suited as benchmark methods for evaluating sparse MKL algorithms.

## 3 An Alternating Best Response Algorithm

In this section, we present an alternating best response algorithm that produces solutions for equation 4. The presence of the $\ell_0$ constraint on $\boldsymbol{\beta}$ makes the problem non-convex and therefore not amenable to commonly used gradient descent ascent style algorithms for minimax optimization. Nevertheless, we show in Sections 3.1 and 3.2 that both the maximization over $\boldsymbol{\alpha}$ for a fixed value of $\boldsymbol{\beta}$ (which we refer to as the $\boldsymbol{\alpha}$-subproblem) and the minimization over $\boldsymbol{\beta}$ for a fixed value of $\boldsymbol{\alpha}$ (which we refer to as the $\boldsymbol{\beta}$-subproblem) can be solved efficiently. In Section 3.4 we explore theoretical convergence and computational complexity properties of our algorithm.

### 3.1 The $\alpha$-Subproblem

For a fixed value $\boldsymbol{\beta} \in \mathbb{R}_+^q$, define the combined kernel $\boldsymbol{K}(\boldsymbol{\beta})$ as $\boldsymbol{K}(\boldsymbol{\beta}) := \sum_{j=1}^q \beta_j \boldsymbol{K}_j$. Then, the $\boldsymbol{\alpha}$-subproblem is given by:

$$\max_{\substack{\boldsymbol{\alpha} \in [0,C]^n \\ \sum_{i=1}^n y_i \alpha_i = 0}} \sum_{i=1}^n \alpha_i - \frac{1}{2}\left(\boldsymbol{y} \circ \boldsymbol{\alpha}\right)^\top \boldsymbol{K}(\boldsymbol{\beta})\left(\boldsymbol{y} \circ \boldsymbol{\alpha}\right). \tag{7}$$

This formulation is equivalent to the standard Kernel SVM dual problem, with the only difference being that the kernel matrix is given by $\boldsymbol{K}(\boldsymbol{\beta})$. As a convex quadratic program, equation 7 can be solved to global optimality using any one of several well-established solvers. In the implementation of our approach, we leverage the LIBSVM solver (Chang & Lin, 2011) which employs the Sequential Minimal Optimization (SMO) algorithm. SMO decomposes the large-scale quadratic program into a series of analytically solvable two-dimensional subproblems. This coordinate descent strategy is both efficient and robust in practice.

### 3.2 The $\beta$-Subproblem

For a fixed value of $\boldsymbol{\alpha} \in \mathbb{R}^n$ that is feasible to equation 7, the $\boldsymbol{\beta}$-subproblem is given by

$$\min_{\boldsymbol{\beta} \in \mathbb{R}_+^q} \quad -\frac{1}{2}(\boldsymbol{y} \circ \boldsymbol{\alpha})^T\left[\sum_{i=1}^q \beta_i \boldsymbol{K}_i\right](\boldsymbol{y} \circ \boldsymbol{\alpha}) + \lambda\|\boldsymbol{\beta}\|_2^2 \quad \text{s.t.} \quad \|\boldsymbol{\beta}\|_1 = 1, \quad \|\boldsymbol{\beta}\|_0 \leq k_0. \tag{8}$$

We define $d_i := \left(\boldsymbol{y} \circ \boldsymbol{\alpha}\right)^\top \boldsymbol{K}_i\left(\boldsymbol{y} \circ \boldsymbol{\alpha}\right)$ for every $i \in [q]$. Given this, we can express the objective function of equation 8 as:

$$f(\boldsymbol{\beta}) = \sum_{i=1}^q \left[-\frac{1}{2}\beta_i d_i + \lambda\beta_i^2\right].$$

For any given $\beta_i$, completing the square shows that we have

$$-\frac{1}{2}\beta_i d_i + \lambda\beta_i^2 = \lambda\left(\beta_i - \frac{d_i}{4\lambda}\right)^2 - \frac{d_i^2}{16\lambda}.$$

Observe that if equation 8 were unconstrained, the optimal solution $\boldsymbol{\beta}^\star$ would be given by $\beta_i^\star = \frac{d_i}{4\lambda}$. Let $\boldsymbol{w} := \frac{\boldsymbol{d}}{4\lambda}$ and define the sets $\Sigma_{k_0} = \{\boldsymbol{\beta} \in \mathbb{R}^q : \|\boldsymbol{\beta}\|_0 \leq k_0\}$ and $\Delta^+ = \{\boldsymbol{\beta} \in \mathbb{R}_+^q : \sum_{j=1}^q \beta_j = 1\}$. It follows that equation 8 has the same optimal solution as:

$$\min_{\boldsymbol{\beta} \in \Sigma_{k_0} \cap \Delta^+} \|\boldsymbol{\beta} - \boldsymbol{w}\|_2^2. \tag{9}$$

In words, solving equation 9 gives the Euclidean projection of the vector $\boldsymbol{w}$ onto the intersection of the sets $\Sigma_{k_0}$ and $\Delta^+$. We obtain the solution to equation 9 by employing the Greedy Selector and Simplex Projector (GSSP) algorithm (Kyrillidis et al., 2013).

GSSP proceeds in three steps:

1. **Top-$k$ truncation $\mathrm{PL}_k$.** Let $S \subset \{1, \ldots, q\}$ index the $k$ largest entries of $\boldsymbol{w}$. Define

$$(\mathrm{PL}_k(\boldsymbol{w}))_i = \begin{cases} w_i, & i \in S, \\ 0, & i \notin S. \end{cases}$$

   This retains only the top $k$ components of $\boldsymbol{w}$.

2. **Simplex projection $P^+$.** Restrict $\boldsymbol{w}$ to the active entries $\{w_i\}_{i \in S}$, sort them into descending order $w_{(1)} \geq w_{(2)} \geq \cdots \geq w_{(k)}$, and compute

$$\tau = \frac{1}{\rho}\Big(\sum_{j=1}^{\rho} w_{(j)} - 1\Big), \quad \rho = \max\Big\{ j : w_{(j)} > \tfrac{1}{j}\big(\sum_{m=1}^{j} w_{(m)} - 1\big) \Big\}.$$

   Then form

$$(P^+(\boldsymbol{w}))_i = \max(w_i - \tau,\, 0) \quad \text{for } i \in S,$$

   and set $(P^+(\boldsymbol{w}))_i = 0$ for $i \notin S$. This enforces nonnegativity and the unit $\ell_1$ norm constraint.

3. **Reconstruction.** The final projected vector is

$$\beta_i = (P^+(\boldsymbol{w}))_i, \quad i = 1, \ldots, q,$$

   which has support $S$ and sums to 1.

By Theorem 1 of Kyrillidis et al. (2013), these steps yield the exact minimizer of equation 9 in quasilinear time.

### 3.3 Alternating Best Response

We now describe our complete alternating best response procedure used to obtain solutions to the sparse multiple kernel learning problem given by equation 4. Our approach iteratively alternates between optimizing the SVM dual variables $\boldsymbol{\alpha}$ and the kernel weights $\boldsymbol{\beta}$. We will see in Section 5 that this heuristic performs well in practice by providing both strong predictive accuracy and kernel interpretability.

Each iteration $t$ of Algorithm 1 consists of two steps. First, we fix $\boldsymbol{\beta} = \boldsymbol{\beta}^{(t-1)}$ and solve the standard SVM dual problem using the kernel matrix $\boldsymbol{K}(\boldsymbol{\beta}^{(t-1)}) = \sum_{j=1}^{q} \beta_j^{(t-1)} \boldsymbol{K_j}$. This subproblem is convex and can be solved to global optimality using either SMO-based solvers (e.g., LIBSVM), or quadratic programming solvers (e.g., Gurobi). This produces an updated vector of dual coefficients $\boldsymbol{\alpha}^{(t)}$. Next, we fix $\boldsymbol{\alpha} = \boldsymbol{\alpha}^{(t)}$ and update the kernel weights $\boldsymbol{\beta}$ by solving equation 9.

**Initialization Strategies.** Before the first iteration, we initialize the kernel weights $\boldsymbol{\beta}^{(0)}$ using one of the following strategies:

- *$k_0$-Sparse Random Initialization:* A subset $S \subseteq \{1, \ldots, q\}$ with $|S| = k_0$ is drawn uniformly at random. We set $\beta_j^{(0)} = 1/k_0$ for $j \in S$ and $\beta_j^{(0)} = 0$ for $j \notin S$.

- *Warm Start:* A feasible vector $\boldsymbol{\beta}^{(0)}$ produced by solving a separate optimization problem (typically a convex lower bound) is used.

**Objective Tracking.** At each iteration, we compute the value of the dual objective associated with the current kernel weights and dual variables. Let $J^{(t)}$ denote the value of the regularized SVM dual objective at iteration $t$, given by

$$J^{(t)} = \sum_{i=1}^{n} \alpha_i^{(t)} - \frac{1}{2} \left( \boldsymbol{y} \circ \boldsymbol{\alpha}^{(t)} \right)^{\top} \boldsymbol{K}^{(t)} \left( \boldsymbol{y} \circ \boldsymbol{\alpha}^{(t)} \right) + \lambda \|\boldsymbol{\beta}^{(t)}\|_2^2,$$

where $\boldsymbol{K}^{(t)} = \sum_{j=1}^{q} \beta_j^{(t)} \boldsymbol{K}_j$ is the kernel matrix induced by the iteration $t$ weights $\boldsymbol{\beta}^{(t)}$, and the final term corresponds to the $\ell_2$ regularization penalty on the kernel weights. This quantity is used to monitor convergence and trigger early termination when progress stalls.

**Stopping Criteria.** The algorithm terminates when either (i) a maximum number of iterations $T$ is reached, or (ii) the absolute improvement in the dual objective falls below a tolerance $\epsilon$ for $M$ consecutive iterations.

---

**Algorithm 1:** Alternating Best Response for Sparse MKL

---

**Data:** $\{(x_i, y_i)\}_{i=1}^{n}$, $\{\boldsymbol{K}_j\}_{j=1}^{q}$, $C$, $\lambda$, $k_0$, $\epsilon$, $M$, $T$
**Result:** $\boldsymbol{\alpha}$, $\boldsymbol{\beta}$
**Init:** choose $\boldsymbol{\beta}^{(0)}$; non_decrease $\leftarrow 0$, obj_best $\leftarrow \infty$
**for** $t = 1, \ldots, T$ **do**                                                              // max $T$ iterations
    $\boldsymbol{K}^{(t)} \leftarrow \sum_j \beta_j^{(t-1)} \boldsymbol{K}_j$
    $\boldsymbol{\alpha}^{(t)} \leftarrow \arg\max$ SVM dual on $\boldsymbol{K}^{(t)}$
    **for** $j = 1, \ldots, q$ **do**
        $d_j \leftarrow (\boldsymbol{y} \circ \boldsymbol{\alpha}^{(t)})^{\top} \boldsymbol{K}_j (\boldsymbol{y} \circ \boldsymbol{\alpha}^{(t)})$
        $w_j \leftarrow d_j / (4\lambda)$
    $\boldsymbol{\beta}^{(t)} \leftarrow \text{GSSP}(\{w_j\}, k_0)$
    Compute objective $J^{(t)}$
    **if** *obj_best* $- J^{(t)} < \epsilon$ **then**
        non_decrease $\leftarrow$ non_decrease $+ 1$
    **else**
        non_decrease $\leftarrow 0$
        obj_best $\leftarrow J^{(t)}$
        Save $\boldsymbol{\alpha}^{(t)}, \boldsymbol{\beta}^{(t)}$
    **if** *non_decrease* $\geq M$ **then**                                        // early stopping
        **break**

---

### 3.4 Algorithm Properties

In this subsection, we briefly explore convergence and complexity properties of Algorithm 1. We defer the proofs of Proposition 1, Proposition 2 and Theorem 3 to Appendix B.

**Proposition 1** *Let $\{\boldsymbol{\alpha}_t, \boldsymbol{\beta}_t\}_{t=1}^{\infty}$ denote the sequence of iterates produced by Algorithm 1 when we ignore the algorithm's stopping criteria. There exist accumulation points for this sequence.*

**Proposition 2** *Let $\{\boldsymbol{\alpha}_t, \boldsymbol{\beta}_t\}_{t=1}^{\infty}$ denote the sequence of iterates produced by Algorithm 1 when we ignore the algorithm's stopping criteria. Suppose the sequence converges. Let $(\boldsymbol{\alpha}^{\star}, \boldsymbol{\beta}^{\star})$ denote the limit of the sequence. Then $(\boldsymbol{\alpha}^{\star}, \boldsymbol{\beta}^{\star})$ is a mutual best response pair.*

It can be seen immediately that Algorithm 1 terminates given any input. It is interesting to consider the asymptotic behavior of the iterates produced by Algorithm 1 in the absence of stopping criteria. Taken

together, Propositions 1 and 2 establish that it is not possible for the iterates of Algorithm 1 to converge to a point that is not a mutual best response. If the iterates produced by Algorithm 1 converge, they must converge to a point $(\boldsymbol{\alpha}^\star, \boldsymbol{\beta}^\star)$ such that $\boldsymbol{\alpha}^\star$ solves equation 7 when $\boldsymbol{\beta} = \boldsymbol{\beta}^\star$ and $\boldsymbol{\beta}^\star$ solves equation 8 when $\boldsymbol{\alpha} = \boldsymbol{\alpha}^\star$.

Propositions 1 and 2 alone do not guarantee convergence for all input. Indeed, it is possible to construct instances where the iterates cycle. It is of interest to understand which conditions guarantee convergence of Algorithm 1. We present one such sufficient condition in Theorem 3. Specifically, Theorem 3 establishes linear convergence to the unique mutual best response pair if the support of the iterates $\boldsymbol{\beta}_t$ stabilizes and the problem hyperparameters satisfies an inequality.

**Theorem 3** *Let $\{\boldsymbol{\alpha}_t, \boldsymbol{\beta}_t\}_{t=1}^\infty$ denote the sequence of iterates produced by Algorithm 1 when we ignore the algorithm's stopping criteria. Suppose there exists an index $T_0$ and a support set $\mathcal{S} \subset [q], |\mathcal{S}| \leq k_0$ such that for all $t \geq T_0$ we have $\text{supp}(\boldsymbol{\beta}_t) = \mathcal{S}$. Suppose further that all kernels are positive definite. Let $\lambda_1(\boldsymbol{K}_i), \lambda_n(\boldsymbol{K}_i)$ denote the largest and smallest eigenvalues of $\boldsymbol{K}_i$ respectively. Suppose the following condition holds:*

$$C^2 n k_0 [\max_j \lambda_1(\boldsymbol{K}_j)]^2 < 2\lambda \min_j \lambda_n(\boldsymbol{K}_j).$$

*Then the sequence $\{\boldsymbol{\alpha}_t, \boldsymbol{\beta}_t\}_{t=1}^\infty$ converges linearly to the unique mutual best response $(\boldsymbol{\alpha}^\star, \boldsymbol{\beta}^\star)$ with rate $\frac{C^2 n k_0 [\max_j \lambda_1(\boldsymbol{K}_j)]^2}{2\lambda \min_j \lambda_n(\boldsymbol{K}_j)}$.*

We conclude this subsection by considering the computational complexity of Algorithm 1. At a given iteration, forming the kernel $\boldsymbol{K}(\boldsymbol{\beta})$ requires $O(k_0 n^2)$ operations (recall that the individual kernels $\boldsymbol{K}_i$ are precomputed). We employ an iterative method to solve the dual SVM problem in $O(nI)$ operations where $I$ denotes the number of iterations required and is empirically known to be higher than linear but less than quadratic in the number of training examples $n$ (Chang & Lin, 2011). Forming the vector $\boldsymbol{d}$ requires $O(qn^2)$ operations and the GSSP algorithm has complexity $O(q \log q)$ being dominated by the cost to sort a $q$-dimensional vector. Thus, the overall complexity of Algorithm 1 is $O(T \cdot [qn^2 + q \log q + nI])$.

## 4 An Exact Reformulation and Convex Relaxations

In this section, we reformulate equation 4 exactly as a mixed integer semidefinite optimization problem and derive several convex relaxations that will be used in Section 5 to both produce lower bounds for solutions returned by Algorithm 1 and to warm start it.

### 4.1 A Mixed Integer Semidefinite Reformulation

We begin by exactly reformulating equation 4 as a mixed integer semidefinite optimization problem. Consider the alpha subproblem given by equation 7. Its dual is given by:

$$
\begin{aligned}
\min_{\eta \in \mathbb{R}, \boldsymbol{\sigma} \in \mathbb{R}_+^n, \boldsymbol{\gamma} \in \mathbb{R}^n} \quad & C \sum_{i=1}^n \sigma_i + \frac{1}{2} \boldsymbol{\gamma}^T \bigg[ \sum_{i=1}^q \beta_i \boldsymbol{K}_i \bigg]^\dagger \boldsymbol{\gamma} \\
\text{s.t.} \quad & 1 - \sigma_i \leq y_i(\eta + \gamma_i) \quad \forall i \in [n], \\
& \bigg( \boldsymbol{I}_n - \big[ \sum_{i=1}^q \beta_i \boldsymbol{K}_i \big] \big[ \sum_{i=1}^q \beta_i \boldsymbol{K}_i \big]^\dagger \bigg) \boldsymbol{\gamma} = 0.
\end{aligned}
\tag{10}
$$

For the interested reader, we present a derivation of equation 10 starting from equation 7 in Appendix C. Since equation 10 contains only affine constraints, Slater's condition is trivially satisfied implying that strong duality holds between equation 7 and equation 10. Notice that equation 10 is a convex quadratic optimization problem since for any $\boldsymbol{\beta}$ that is feasible in equation 4, the matrix $\big[ \sum_{i=1}^q \beta_i \boldsymbol{K}_i \big]^\dagger$ will be positive semidefinite. We seek to leverage equation 10 to replace the inner maximization problem in equation 4 which would yield a single minimization problem.

Directly substituting the inner maximization of equation 4 with equation 10 is not desirable since $\boldsymbol{\beta}$ would be an optimization variable that appears within matrix pseudo inverse terms and is multiplied by $\boldsymbol{\gamma}$, another optimization variable, in the objective function and in the last constraint. Consequently, we proceed by introducing a semidefinite constraint that allows equation 10 to be reformulated as a semidefinite optimization problem without products between $\boldsymbol{\gamma}$ and $\boldsymbol{\beta}$ or matrix pseudo inverse terms. Consider the optimization problem given by:

$$
\begin{aligned}
\min_{\eta,\theta\in\mathbb{R},\boldsymbol{\sigma}\in\mathbb{R}^n_+,\boldsymbol{\gamma}\in\mathbb{R}^n} \quad & C\sum_{i=1}^{n}\sigma_i + \frac{1}{2}\theta \\
\text{s.t.} \quad & 1-\sigma_i \le y_i(\eta+\gamma_i) \quad \forall\, i\in[n], \\
& \begin{pmatrix} \theta & \boldsymbol{\gamma}^T \\ \boldsymbol{\gamma} & \left[\sum_{i=1}^{q}\beta_i\boldsymbol{K}_i\right] \end{pmatrix} \succeq 0.
\end{aligned}
\tag{11}
$$

It follows from the Generalized Schur Complement Lemma (see Boyd et al. (1994), Equation 2.41) that equation 10 and equation 11 achieve the same optimal objective value. We state this equivalence formally in Proposition 4 (proof deferred to Appendix D).

**Proposition 4** *Equation 11 is a valid reformulation of equation 10.*

We can now obtain a more tractable reformulation of equation 4 by substituting its inner maximization problem with the equivalent minimization equation 11. Consider the optimization problem given by:

$$
\begin{aligned}
\min_{\substack{\eta,\theta\in\mathbb{R},\boldsymbol{\sigma}\in\mathbb{R}^n_+, \\ \boldsymbol{\gamma}\in\mathbb{R}^n,\boldsymbol{\beta}\in\mathbb{R}^q_+}} \quad & C\sum_{i=1}^{n}\sigma_i + \frac{1}{2}\theta + \lambda\|\boldsymbol{\beta}\|_2^2, \\
\text{s.t.} \quad & 1-\sigma_i \le y_i(\eta+\gamma_i) \quad \forall\, i\in[n], \\
& \begin{pmatrix} \theta & \boldsymbol{\gamma}^T \\ \boldsymbol{\gamma} & \left[\sum_{i=1}^{q}\beta_i\boldsymbol{K}_i\right] \end{pmatrix} \succeq 0, \\
& \sum_{i=1}^{q}\beta_i = 1, \|\boldsymbol{\beta}\|_0 \le k_0.
\end{aligned}
\tag{12}
$$

equation 12 consists of a non-convex minimization problem where the nonconvexity is entirely captured by the cardinality constraint $\|\boldsymbol{\beta}\|_0 \le k_0$. We further reformulate equation 12 by introducing binary variables $\boldsymbol{z}\in\{0,1\}^q$ and imposing the nonlinear constraint $\beta_i = z_i\beta_i$ to model the sparsity pattern of $\boldsymbol{\beta}$. Explicitly, we characterize the set of $q$-dimensional vectors with cardinality at most $k_0$ using the following equivalence:

$$
\{\boldsymbol{\beta}\in\mathbb{R}^q : \|\boldsymbol{\beta}\|_0 \le k_0\} = \{\boldsymbol{\beta}\in\mathbb{R}^q : \exists\, \boldsymbol{z}\in\{0,1\}^q, \sum_{i=1}^{q}z_i \le k_0, \beta_i = z_i\beta_i \,\forall\, i\}.
$$

As $\boldsymbol{z}$ and $\boldsymbol{\beta}$ will both be optimization variables in our relaxation, the constraints $\beta_i = z_i\beta_i$ are complicating as they are non-convex in the decision variables. Accordingly, we invoke the perspective reformulation (Günlük & Linderoth, 2012) to model these constraints in a convex manner. Specifically, we introduce a variable $\boldsymbol{\omega}\in\mathbb{R}^n$ where $\omega_i$ models $\beta_i^2$ and we introduce constraints $\omega_i z_i \ge \beta_i^2$ which are second-order cone representable. This results in the following exact reformulation of equation 4 and equation 12:

$$\min_{\substack{\eta,\theta\in\mathbb{R},\boldsymbol{\sigma}\in\mathbb{R}_+^n,\boldsymbol{\gamma}\in\mathbb{R}^n,\\ \boldsymbol{\beta},\boldsymbol{\omega}\in\mathbb{R}_+^q,\boldsymbol{z}\in\{0,1\}^q}} \quad C\sum_{i=1}^{n}\sigma_i + \frac{1}{2}\theta + \lambda\sum_{i=1}^{q}\omega_i,$$

$$\text{s.t.} \quad 1-\sigma_i \le y_i(\eta+\gamma_i) \quad \forall\,i\in[n],$$

$$\begin{pmatrix}\theta & \boldsymbol{\gamma}^T \\ \boldsymbol{\gamma} & \left[\sum_{i=1}^{q}\beta_i\boldsymbol{K}_i\right]\end{pmatrix} \succeq 0, \tag{13}$$

$$\sum_{i=1}^{q}\beta_i = 1, \quad \sum_{i=1}^{q}z_i \le k_0, \quad \beta_i^2 \le z_i\omega_i \quad \forall\,i\in[q].$$

## 4.2 Positive Semidefinite Cone Relaxations

Equation 13 is an exact mixed integer semidefinite optimization reformulation of equation 4 where the problem's non-convexity is entirely captured through the binary condition $\boldsymbol{z}\in\{0,1\}^q$. We now obtain a semidefinite relaxation by solving equation 13 with $\boldsymbol{z}\in\text{conv}(\{0,1\}^q)=[0,1]^q$. This yields the following convex optimization problem, which we refer to as the *Full SDP* relaxation:

$$\min_{\substack{\eta,\theta\in\mathbb{R},\boldsymbol{\sigma}\in\mathbb{R}_+^n,\boldsymbol{\gamma}\in\mathbb{R}^n,\\ \boldsymbol{\beta},\boldsymbol{\omega}\in\mathbb{R}_+^q,\boldsymbol{z}\in[0,1]^q}} \quad C\sum_{i=1}^{n}\sigma_i + \frac{1}{2}\theta + \lambda\sum_{i=1}^{q}\omega_i,$$

$$\text{s.t.} \quad 1-\sigma_i \le y_i(\eta+\gamma_i) \quad \forall\,i\in[n],$$

$$\begin{pmatrix}\theta & \boldsymbol{\gamma}^T \\ \boldsymbol{\gamma} & \left[\sum_{i=1}^{q}\beta_i\boldsymbol{K}_i\right]\end{pmatrix} \succeq 0, \tag{14}$$

$$\sum_{i=1}^{q}\beta_i = 1, \quad \sum_{i=1}^{q}z_i \le k_0, \quad \beta_i^2 \le z_i\omega_i \quad \forall\,i\in[q].$$

**Theorem 5** *Equation 14 is a valid convex relaxation of equation 4.*

We defer the proof of Theorem 5 to Appendix D.

In Section 5, we use equation 14 to produce lower bounds for feasible solutions returned by Algorithm 1. Note that Algorithm 1 and equation 14 can be leveraged to design a custom branch and bound algorithm in the sense of Little (1966); Land & Doig (2010) to solve equation 4 to global optimality. Such an approach would involve constructing an enumeration tree that branches on the entries of the vector $\boldsymbol{z}$. Each node in the enumeration tree would be defined by two disjoint collections of indices $\mathcal{I}_0,\mathcal{I}_1\subseteq[q]$ where $\mathcal{I}_i$ denotes indices of $\boldsymbol{z}$ constrained to take value $i$. At a given node, feasible solutions and lower bounds would be computed using slightly modified versions of Algorithm 1 and equation 14 respectively.

As a brief aside, we note that in the special case where all of the input kernels $\{\boldsymbol{K}_i\}_{i=1}^{q}$ are simultaneously diagonalizable, equation 14 can be reduced to a second order cone problem. Explicitly, consider the following optimization problem:

$$\min_{\substack{\eta,\theta\in\mathbb{R},\boldsymbol{\sigma}\in\mathbb{R}_+^n,\boldsymbol{\gamma},\boldsymbol{\tau}\in\mathbb{R}^n,\\ \boldsymbol{\beta},\boldsymbol{\omega}\in\mathbb{R}_+^q,\boldsymbol{z}\in[0,1]^q}} \quad C\sum_{i=1}^{n}\sigma_i + \frac{1}{2}\theta + \lambda\sum_{i=1}^{q}\omega_i,$$

$$\text{s.t.} \quad 1-\sigma_i \le y_i(\eta+\gamma_i) \quad \forall\,i\in[n],$$

$$\tau_j \cdot \sum_{i=1}^{q}\beta_i[\boldsymbol{D}_i]_{jj} \ge (\boldsymbol{\gamma}^T\boldsymbol{u}_j)^2 \quad \forall\,j\in[n], \tag{15}$$

$$\theta \ge \sum_{j=1}^{n}\tau_j, \quad \sum_{i=1}^{q}\beta_i = 1, \quad \sum_{i=1}^{q}z_i \le k_0, \quad \beta_i^2 \le z_i\omega_i \quad \forall\,i\in[q],$$

where $\boldsymbol{K}_i = \boldsymbol{U}\boldsymbol{D}_i\boldsymbol{U}^T$ for all $i \in [q]$ are spectral decompositions of the candidate kernels which are assumed to be simultaneously diagonalizable. Note that $\boldsymbol{u}_j \in \mathbb{R}^n$ denotes the $j^{th}$ column of $\boldsymbol{U} \in \mathbb{R}^{n \times n}$. We now have the following result (proof deferred to Appendix D):

**Theorem 6** *If the candidate kernels $\{\boldsymbol{K}_i\}_{i=1}^q$ are simultaneously diagonalizable, meaning that we have $\boldsymbol{K}_i = \boldsymbol{U}\boldsymbol{D}_i\boldsymbol{U}^T$ for all $i \in [q]$ where $D_i \in \mathbb{R}^{n \times n}$ are diagonal matrices and $\boldsymbol{U} \in \mathbb{R}^{n \times n}$ satisfies $\boldsymbol{U}\boldsymbol{U}^T = \boldsymbol{U}^T\boldsymbol{U} = \boldsymbol{I}_n$, then equation 14 is equivalent to equation 15.*

We stress that the assumption in Theorem 6 of simultaneously diagonalizable kernels is a strong assumption that in general will not hold in practice.

### 4.2.1 Sparse Positive Semidefinite Cone Approximations

We will see in Section 5 that equation 14 empirically produces strong lower bounds. However, solving equation 14 becomes computationally difficult as the number of training examples $n$ grows because even the most efficient interior point solvers for semidefinite optimization problems exhibit poor scaling with the size of the positive semidefinite constraint which in the case of equation 14 has dimension $(n+1) \times (n+1)$. In order to produce lower bounds efficiently for large problem instances, we are interested in developing approximations to equation 14 that can be computed more efficiently as the problem dimension scales.

To this end, we leverage a common approximation approach that involves relaxing the global positive semidefinite constraint and instead, for a given value $\ell \in \mathbb{N}$, constraining all $\ell \times \ell$ principal submatrices to be positive semidefinite. This technique is often called the sparse SDP relaxation and is known to have desirable approximation properties in both theory and practice (Blekherman et al., 2022; Song & Parrilo, 2023). Recall that $\mathbb{S}_+^n$ denotes the set of all $n \times n$ real symmetric positive semidefinite matrices and let $\mathbb{S}_\ell^n$ denote the set of all $n \times n$ real symmetric matrices where all $\ell \times \ell$ principal submatrices are positive semidefinite. It is clear that for any $\ell \in \mathbb{N}$, we have $\mathbb{S}_+^n \subseteq \mathbb{S}_{\ell+1}^n \subseteq \mathbb{S}_\ell^n$.

In general, for a matrix $\boldsymbol{X} \in \mathbb{R}^{(n+1) \times (n+1)}$, substituting the constraint $\boldsymbol{X} \in \mathbb{S}_+^{n+1}$ by $\boldsymbol{X} \in \mathbb{S}_\ell^{n+1}$ would involve replacing a $(n+1) \times (n+1)$ semidefinite constraint with $\binom{n+1}{\ell}$ $\ell \times \ell$ semidefinite constraints. However, recall that the $n \times n$ sub block $\sum_{i=1}^q \beta_i \boldsymbol{K}_i$ of the $(n+1) \times (n+1)$ matrix involved in the semidefinite constraint of equation 14 is itself guaranteed to be positive semidefinite for any feasible $\boldsymbol{\beta}$. Accordingly, any principal submatrix that is entirely contained within this sub block is guaranteed to be positive semidefinite which implies we need only concern ourselves with principal submatrices that involve the first row and column when defining our approximation. Thus, letting $\boldsymbol{X} = \begin{pmatrix} \theta & \boldsymbol{\gamma}^T \\ \boldsymbol{\gamma} & \left[\sum_{i=1}^q \beta_i \boldsymbol{K}_i\right] \end{pmatrix}$, imposing the constraint $\boldsymbol{X} \in \mathbb{S}_\ell^{n+1}$ requires only $\binom{n}{\ell-1}$ $\ell \times \ell$ semidefinite constraints rather than $\binom{n+1}{\ell}$ $\ell \times \ell$ many. This equates to using $\binom{n}{\ell-1} \times (\frac{n}{\ell} - 1)$ fewer constraints.

As we report results for $\ell = 2$ and $\ell = 3$ in Section 5, we explicitly state the corresponding relaxations below. For $\ell = 2$, the approximation to equation 14 is given by:

$$
\begin{aligned}
\min_{\substack{\eta, \theta \in \mathbb{R}, \boldsymbol{\sigma} \in \mathbb{R}_+^n, \boldsymbol{\gamma} \in \mathbb{R}^n, \\ \boldsymbol{\beta}, \boldsymbol{\omega} \in \mathbb{R}_+^q, \boldsymbol{z} \in [0,1]^q}} \quad & C \sum_{i=1}^n \sigma_i + \frac{1}{2}\theta + \lambda \sum_{i=1}^q \omega_i, \\
\text{s.t.} \quad & 1 - \sigma_i \leq y_i(\eta + \gamma_i) \quad \forall\, i \in [n], \\
& \theta \cdot \sum_{i=1}^q \beta_i [\boldsymbol{K}_i]_{jj} \geq \gamma_j^2 \quad \forall\, j \in [n], \\
& \sum_{i=1}^q \beta_i = 1, \quad \sum_{i=1}^q z_i \leq k_0, \quad \beta_i^2 \leq z_i \omega_i \quad \forall\, i \in [q].
\end{aligned}
\tag{16}
$$

Equation 16 is a second order cone relaxation of equation 14. In Section 5, we refer to this relaxation as *Second Order Cone Relaxation with basis vectors*. Moreover, we can strengthen this second order cone relaxation by randomly sampling unit vectors $\boldsymbol{x} \in \mathbb{R}^n$ and imposing the additional second order cone representable

constraints given by $\theta \cdot \sum_{i=1}^{q} \beta_i(\boldsymbol{x}^T \boldsymbol{K}_i \boldsymbol{x}) \geq (\boldsymbol{x}^T \boldsymbol{\gamma})^2$ for each sampled unit vector $\boldsymbol{x}$. To see why these constraints are valid for equation 14, notice that by the schur complement lemma the semidefinite constraint in equation 14 implies that the matrix $\sum_{i=1}^{q} \beta_i \boldsymbol{K}_i - \frac{1}{\theta} \boldsymbol{\gamma}\boldsymbol{\gamma}^T$ must also be positive semidefinite. This condition can equivalently be expressed as $\theta \cdot \sum_{i=1}^{q} \beta_i \boldsymbol{K}_i \succeq \boldsymbol{\gamma}\boldsymbol{\gamma}^T$. Thus, for any vector $\boldsymbol{x}$ (whether a unit vector or not), we have $\theta \cdot \sum_{i=1}^{q} \beta_i(\boldsymbol{x}^T \boldsymbol{K}_i \boldsymbol{x}) \geq (\boldsymbol{x}^T \boldsymbol{\gamma})^2$. Notice that if we generate these constraints letting $\boldsymbol{x}$ be the standard basis vectors, we obtain the second order cone constraints in equation 16. Explicitly, let $\mathcal{L}$ denote a collection of randomly sampled unit vectors. Consider the optimization problem given by:

$$
\begin{aligned}
\min_{\substack{\eta,\theta\in\mathbb{R}, \boldsymbol{\sigma}\in\mathbb{R}_+^n, \boldsymbol{\gamma}\in\mathbb{R}^n, \\ \boldsymbol{\beta},\boldsymbol{\omega}\in\mathbb{R}_+^q, \boldsymbol{z}\in[0,1]^q}} \quad & C\sum_{i=1}^{n}\sigma_i + \frac{1}{2}\theta + \lambda\sum_{i=1}^{q}\omega_i, \\
\text{s.t.} \quad & 1-\sigma_i \leq y_i(\eta+\gamma_i) \quad \forall\, i\in[n], \\
& \theta \cdot \sum_{i=1}^{q}\beta_i \boldsymbol{x}^T \boldsymbol{K}_i \boldsymbol{x} \geq (\boldsymbol{x}^T\boldsymbol{\gamma})^2 \quad \forall\, \boldsymbol{x}\in\mathcal{L}\cup\{\boldsymbol{e}_j\}_{j=1}^{n}, \\
& \sum_{i=1}^{q}\beta_i = 1, \quad \sum_{i=1}^{q}z_i \leq k_0, \quad \beta_i^2 \leq z_i\omega_i \quad \forall\, i\in[q].
\end{aligned}
\tag{17}
$$

Equation 17 is a second order cone relaxation of equation 14. In Section 5, we refer to this relaxation as *Second Order Cone Relaxation with randomized vectors*. Moreover it should be clear that equation 17 is a stronger relaxation than equation 16 since the feasible set of equation 17 is a subset of that of equation 16. Moreover, equation 17 becomes an increasingly strong approximation to equation 14 as the cardinality of $\mathcal{L}$ increases. We will explore this further in Section 5. The final relaxation we introduce is the sparse SDP relaxation with $\ell = 3$ for equation 14 which is given by:

$$
\begin{aligned}
\min_{\substack{\eta,\theta\in\mathbb{R}, \boldsymbol{\sigma}\in\mathbb{R}_+^n, \boldsymbol{\gamma}\in\mathbb{R}^n, \\ \boldsymbol{\beta},\boldsymbol{\omega}\in\mathbb{R}_+^q, \boldsymbol{z}\in[0,1]^q}} \quad & C\sum_{i=1}^{n}\sigma_i + \frac{1}{2}\theta + \lambda\sum_{i=1}^{q}\omega_i, \\
\text{s.t.} \quad & 1-\sigma_i \leq y_i(\eta+\gamma_i) \quad \forall\, i\in[n], \\
& \begin{pmatrix} \theta & \gamma_j & \gamma_k \\ \gamma_j & \sum_{i=1}^{q}\beta_i[\boldsymbol{K}_i]_{jj} & \sum_{i=1}^{q}\beta_i[\boldsymbol{K}_i]_{jk} \\ \gamma_k & \sum_{i=1}^{q}\beta_i[\boldsymbol{K}_i]_{kj} & \sum_{i=1}^{q}\beta_i[\boldsymbol{K}_i]_{kk} \end{pmatrix} \succeq 0 \quad \forall\, 1\leq j,k\leq n, j\neq k, \\
& \sum_{i=1}^{q}\beta_i = 1, \quad \sum_{i=1}^{q}z_i \leq k_0, \quad \beta_i^2 \leq z_i\omega_i \quad \forall\, i\in[q].
\end{aligned}
\tag{18}
$$

In Section 5, we refer to this relaxation with $\ell = 3$ as *3x3-SDP* relaxation.

## 5 Computational Results

In this section, we evaluate the performance of Algorithm 1 and the relaxations introduced in Section 4. We benchmark against open source methods implemented in `MKLpy` library (Lauriola & Aiolli, 2020) that we introduced in Section 2.2. All experiments were run using Julia 1.10.1 and Python 3.10.14. Semidefinite programs were solved with `MOSEK 10.1.31`, while all other optimization relied on Julia's `LIBSVM` v0.8.0 and `MKLpy` 0.6. We perform experiments on MIT's Supercloud Cluster (Reuther et al., 2018), which hosts Intel Xeon Platinum 8260 processors. To bridge the gap between theory and practice, we have made our code freely available at `https://github.com/iglesiascaio/SparseMKL`.

We seek to answer four questions:

**Q1** *Predictive accuracy and runtime.* How does Algorithm 1 compare to other widely used open-source MKL solvers—EasyMKL, AverageMKL, and CKA—implemented in `MKLpy` (Lauriola & Aiolli, 2020), in terms of prediction accuracy and computational efficiency?

**Q2** *Warm starts.* Does initializing Algorithm 1 with solutions from our convex relaxations enhance the quality of the returned solution?

**Q3** *Optimality.* How close is the objective value achieved by the output of Algorithm 1 to the lower bounds produced by the SDP relaxations of Section 4?

**Q4** *Scalability.* How does Algorithm 1 as the number of kernels $q$ increases?

We address Q1–Q4 in the following sections.

### 5.1 Experimental design

**Data sets.** We evaluate all methods on ten benchmark binary classification tasks drawn from the UCI Machine Learning Repository (Dua & Graff, 2017). Seven of these—BREASTCANCER (Wisconsin Diagnostic), IONOSPHERE, SPAMBASE, BANKNOTE, HABERMAN, MAMMOGRAPHIC, and PARKINSONS—are provided in their original binary form. The remaining three—IRIS, WINE, and HEART (Cleveland)—are originally multiclass and were converted into binary tasks.

Each data set was parsed into numeric features, with any categorical variables one-hot encoded while omitting one dimension to avoid collinearity. To ensure reproducibility, we fixed the random seed, shuffled the rows, and split 80% of the examples into a training set and the remaining 20% into a test set. Numeric features were standardized to zero mean and unit variance using statistics computed solely on the training set; the same centering and scaling parameters were then applied to the test set. Table 1 summarizes, for each task, the original number of examples, the train set size $n_{\text{tr}}$, the test set size $n_{\text{te}}$, and the proportion of positive labels.

Table 1: Summary of binary classification tasks used in our experiments. The table reports, for each dataset, the number of examples $n$, the training size $n_{\text{tr}}$, the test size $n_{\text{te}}$, and the proportion of positive labels.

| Dataset | $n$ | $n_{\text{tr}}$ | $n_{\text{te}}$ | % Positive (+1) |
|---|---|---|---|---|
| IRIS | 150 | 120 | 30 | 33.3% |
| WINE | 178 | 142 | 36 | 33.1% |
| BREASTCANCER | 569 | 455 | 114 | 37.3% |
| IONOSPHERE | 351 | 280 | 71 | 64.1% |
| SPAMBASE | 4601 | 3680 | 921 | 39.4% |
| BANKNOTE | 1372 | 1097 | 275 | 44.5% |
| HEART | 303 | 242 | 61 | 45.9% |
| HABERMAN | 306 | 244 | 62 | 73.5% |
| MAMMOGRAPHIC | 961 | 768 | 193 | 46.3% |
| PARKINSONS | 195 | 156 | 39 | 75.4% |

**Set of Base Kernels.** All methods are evaluated on the same predefined set of ten base kernels. Our goal in assembling this collection was to include a diverse yet representative sample of the similarity measures introduced in Appendix A, so as to provide each model with flexibility across different feature space geometries.

Concretely, our set comprises a linear kernel (to capture simple inner-product structure), three polynomial kernels of degree $d \in \{2, 3, 5\}$ with offset $c_0 = 1$ and $\alpha = 0.01$, and three Gaussian radial-basis-function (RBF) kernels with parameters $\gamma \in \{0.5, 0.3, 0.1\}$. We further include two sigmoid kernels, each with offset $c_0 = 1$ and slopes $\gamma \in \{0.5, 0.7\}$, and a single Laplacian kernel with $\gamma = 0.3$.

Each kernel matrix is precomputed once and then symmetrized via the transformation $\boldsymbol{K} \leftarrow \frac{1}{2}(\boldsymbol{K} + \boldsymbol{K}^\top)$ to guard against numerical asymmetry and guarantee positive semidefiniteness. Finally, we add $10^{-6} \cdot \boldsymbol{I}$ to every kernel matrix to improve conditioning and ensure stable solver performance.

**Hyperparameter tuning.** For Algorithm 1, we perform 10-fold cross-validation over the training set to select the regularization and sparsity parameters from the grid

$$C \in \{5, 10, 50, 100\}, \quad \lambda \in \{0.01, 0.1, 1, 10, 100\}, \quad k_0 \in \{1, 2, 3, 4, 5\}.$$

The same folds are reused to tune EasyMKL's margin-variance trade-off parameter $\lambda_{\text{Easy}}$ over 25 values in $[10^{-4}, 1]$. AverageMKL and CKA require no parameter tuning. Once the best hyperparameters are identified, each model is retrained on the full training set and evaluated on the withheld test set.

**Performance metrics.** Classification quality is measured by test accuracy (Table 2); solution quality by the MKL objective value (Table 6); optimality by the percentage gap with respect to the tightest available lower bound (Table 7); and computational burden by the training time (Table 4).

## 5.2 Predictive Accuracy and Runtime Compared to Open-Source MKL Solvers (Q1)

We begin our computational evaluation by assessing the predictive performance and runtime of Algorithm 1 in comparison to existing open-source MKL solvers.

Table 2 reports the test set accuracy achieved by EasyMKL, AverageMKL, CKA, and Algorithm 1 with random initialization. We find that Algorithm 1 matches or outperforms the best open-source baseline on all tasks: it strictly improves over the strongest competitor in eight cases and ties in the remaining two. The largest absolute gains occur on IONOSPHERE (+7.1 percentage points), HEART (+6.5 points), and PARKINSONS (+7.6 points), with additional consistent improvements observed on WINE (+2.8 points), BREASTCANCER (+3.6 points), HABERMAN (+1.6 points), and MAMMOGRAPHIC (+3.7 points). On the perfectly separable tasks IRIS and BANKNOTE, Algorithm 1 attains 100% test accuracy, matching the best-performing solvers. Averaged across datasets, Algorithm 1 outperforms the strongest baseline by 3.34 percentage points.

| Dataset | Train Size | EasyMKL | AverageMKL | CKA | Algorithm 1 |
|---|---|---|---|---|---|
| iris | 120 | **100.0** | **100.0** | 96.7 | **100.0** |
| wine | 142 | 97.2 | 97.2 | 91.7 | **100.0** |
| breastcancer | 455 | 93.0 | 92.1 | 94.7 | **98.3** |
| ionosphere | 280 | 73.2 | 74.6 | 85.9 | **93.0** |
| spambase | 3680 | 90.4 | 87.6 | 81.0 | **90.9** |
| banknote | 1097 | **100.0** | **100.0** | 85.1 | **100.0** |
| heart | 242 | 85.2 | 85.2 | 86.9 | **93.4** |
| haberman | 244 | 61.3 | 62.9 | 66.1 | **67.7** |
| mammographic | 768 | 80.8 | 79.3 | 75.1 | **84.5** |
| parkinsons | 156 | 82.1 | 82.1 | 74.4 | **89.7** |

Table 2: Test accuracy (%) of EasyMKL, AverageMKL, CKA, and Algorithm 1 with random initialization.

Crucially, these gains are achieved while selecting sparse kernel combinations as illustrated by Table 3. In contrast, AverageMKL uniformly assigns nonzero weight to all kernels, EasyMKL encourages sparsity to some extent via regularization but does not allow explicit control over the sparsity level; and CKA typically yields dense combinations of all kernels. We stress that the randomly initialized variant of Algorithm 1 achieves these improvements without solving any semidefinite programming relaxations and in only a modest number of alternating best response steps. Consequently, the runtime of Algorithm 1 is competitive with standard MKL solvers as illustrated in Table 4.

| Dataset | EasyMKL | AverageMKL | CKA | Algorithm 1 |
|---|---|---|---|---|
| iris | 10 | 10 | 9 | **1** |
| wine | 4 | 10 | 7 | **2** |
| breastcancer | 3 | 10 | 7 | **2** |
| ionosphere | 3 | 10 | 6 | **1** |
| spambase | 10 | 10 | 7 | **5** |
| banknote | 10 | 10 | 7 | **1** |
| heart | **3** | 10 | 7 | 4 |
| haberman | 8 | 10 | 8 | **1** |
| mammographic | 7 | 10 | 7 | **1** |
| parkinsons | **2** | 10 | 7 | **2** |

Table 3: Number of non-zero coefficients ($|\beta_j| > 10^{-3}$) for each algorithm on every dataset.

| Dataset | Train Size | EasyMKL | AverageMKL | CKA | Algorithm 1 |
|---|---|---|---|---|---|
| iris | 120 | 0.030 | **0.002** | 0.026 | 0.022 |
| wine | 142 | 0.045 | **0.002** | 0.029 | 0.005 |
| breastcancer | 455 | 0.244 | **0.004** | 0.061 | 0.110 |
| ionosphere | 280 | 0.072 | **0.003** | 0.036 | 0.070 |
| spambase | 3680 | 33.671 | **0.631** | 8.859 | 8.330 |
| banknote | 1097 | 0.337 | **0.006** | 0.301 | 0.519 |
| heart | 242 | 0.062 | **0.003** | 0.035 | 0.188 |
| haberman | 244 | 0.029 | 1.907 | 0.033 | **0.025** |
| mammographic | 768 | 0.228 | 13.027 | **0.125** | 0.436 |
| parkinsons | 156 | 0.038 | **0.002** | 0.028 | 0.245 |

Table 4: Training time (seconds) of EasyMKL, AverageMKL, CKA, and Algorithm 1 with random initialization. The fastest runtime in each row is bolded.

It should come as no surprise that AverageMKL has the fastest runtime in eight of the ten benchmarks since this method just takes a naive average of the input kernels. Among the three weight-learning algorithms, performance is more nuanced. CKA posts the shortest times on a majority of datasets, yet its advantage over Algorithm 1 seldom exceeds a few hundredths of a second and disappears entirely on the largest workload: on SPAMBASE (3,680 data points) Algorithm 1 is the fastest learner, running 6% faster than CKA and over four times faster than EasyMKL. Algorithm 1 also takes the lead on IRIS, WINE, and HABERMAN. EasyMKL offers savings relative to Algorithm 1 on some mid-sized tasks (e.g., PARKINSONS, MAMMOGRAPHIC), but it is the slowest option on the largest problem.

These results highlight Algorithm 1's favorable trade-off between predictive accuracy, model sparsity, and computational efficiency in the case of random initialization. This variant of Algorithm 1 suffices to achieve state-of-the-art performance across a broad range of benchmarks, offering strong predictive accuracy, interpretable sparsity, and practical training times.

## 5.3 Effect of Warm Starts (Q2)

We next evaluate the practical benefits of initializing Algorithm 1 with solutions of SDP relaxations. We consider three variants:

- **Random:** Employ the $k_0$-sparse random initialization technique described in Section 3.3;

- $3 \times 3$**-SDP:** Solve our sparse semidefinite relaxation for $\ell = 3$, given by equation 18. Select the $k_0$ largest entries of $\boldsymbol{z}$ as the support of $\boldsymbol{\beta}$ and scale $\boldsymbol{\beta}$ (restricted to these entries) to have unit $\ell_1$ norm;

- **Full SDP:** Solve the dense semidefinite relaxation given by equation 14. Select the $k_0$ largest entries of $\boldsymbol{z}$ as the support of $\boldsymbol{\beta}$ and scale $\boldsymbol{\beta}$ (restricted to these entries) to have unit $\ell_1$ norm.

Table 5 reports test set accuracy for these variants. Warm starts occasionally yield improvements, but their effect is heterogeneous across datasets. In some cases, warm initialization produces significant gains: on PARKINSONS, the full-warm start improves accuracy from 89.7% to 94.9%, and on IONOSPHERE, the $3\times3$-warm start increases accuracy from 93.0% to 94.4%.

| Dataset | Train Size | Best Baseline (max of benchmarks) | Algorithm 1 Variants | | |
|---|---|---|---|---|---|
| | | | (random) | (warm 3×3) | (warm full) |
| iris | 120 | **100.0** | **100.0** | **100.0** | **100.0** |
| wine | 142 | 97.2 | **100.0** | **100.0** | **100.0** |
| breastcancer | 455 | 94.7 | **98.3** | 97.4 | – |
| ionosphere | 280 | 85.9 | 93.0 | **94.4** | 90.1 |
| spambase | 3680 | 90.4 | 90.9 | **93.9**$^\dagger$ | – |
| banknote | 1097 | **100.0** | **100.0** | **100.0** | – |
| heart | 242 | 86.9 | **93.4** | **93.4** | **93.4** |
| haberman | 244 | 66.1 | **67.7** | 66.1 | 66.1 |
| mammographic | 768 | 80.8 | **84.5** | **84.5** | – |
| parkinsons | 156 | 82.1 | 89.7 | 89.7 | **94.9** |

Table 5: Best baseline accuracy (highest of EasyMKL, AverageMKL, CKA) versus Algorithm 1 variants. "–" indicates that results were unavailable due to memory limits. $^\dagger$For SPAMBASE, the $3\times3$ warm start uses the randomized SOC relaxation due to memory limits.

In other cases, however, the benefits are marginal or nonexistent: on HABERMAN, for instance, the random start already achieves 67.7% accuracy, and both warm starts slightly reduce performance to 66.1%. Overall, while warm starts sometimes improve the quality of the solution, they do not consistently dominate the random start variant across all benchmarks. Averaged across datasets, the best warm-started variant of Algorithm 1 improves upon the strongest baseline by 4.05 percentage points, where for each dataset the improvement is defined as the difference between the maximum result of the full SDP and $3\times3$ minor warm-starts and the best baseline. This exceeds the 3.34-point gain of the random-start variant.

It is important to note that for SPAMBASE, the $3\times3$-SDP warm entry in Table 5 does *not* correspond to the true $3\times3$-SDP, but instead uses a randomized second-order cone (SOC) relaxation given by equation 17. This substitution was necessary because both the full SDP and the $3\times3$ SDP exceeded available memory on this large dataset. Interestingly, even though the SOC relaxation provides an extremely loose lower bound—far from the heuristic objective value (0.2 vs. over 700 as shown in Table 6 in Section 5.4)—warm-starting from the SOC solution still improves the test accuracy substantially, from 90.9% to 93.9%. This highlights that even relatively weak relaxations can offer valuable guidance for initialization when solving large-scale SMKL problems with Algorithm 1.

While warm starts can help improve predictive performance, they impose substantial computational burdens. Both relaxations become impractical for large problems because of the poor memory scalability exhibited by interior point methods when used to solve semidefinite programs. In contrast, the randomly initialized variant of Algorithm 1 requires only a modest number of alternating best response steps, scaling efficiently even for datasets with thousands of examples. Despite its simplicity, the random start variant achieves competitive or superior predictive performance across all benchmarks without relying on solving costly semidefinite programs.

### 5.4 Optimality certification via SDP relaxations (Q3)

Using the semidefinite relaxations introduced in Section 4, we can certify how close the solutions returned by Algorithm 1 are to a global optimum. We consider three levels of relaxation:

- **Second Order Cone (SOC):** Solve our sparse semidefinite relaxation with $\ell = 2$ with standard basis and randomized vectors, given by equation 17;

- $3 \times 3$**-SDP:** Solve our sparse semidefinite relaxation for $\ell = 3$, given by equation 18;

- **Full SDP:** Solve the dense semidefinite relaxation give by equation 14.

Each of these yields a *lower bound* $f_{\mathrm{LB}}$ on the true optimal MKL objective. Given any SMKL feasible solution $(\hat{\boldsymbol{\beta}}, \hat{\boldsymbol{\alpha}})$, we compute its objective value $F(\hat{\boldsymbol{\beta}}, \hat{\boldsymbol{\alpha}})$ and report the *relative optimality gap*

$$\mathrm{Gap}(\hat{\boldsymbol{\beta}}, \hat{\boldsymbol{\alpha}}) \;=\; \frac{F(\hat{\boldsymbol{\beta}}, \hat{\boldsymbol{\alpha}}) - f_{\mathrm{LB}}}{F(\hat{\boldsymbol{\beta}}, \hat{\boldsymbol{\alpha}})} \times 100\%,$$

which is an upper bound on how far $(\hat{\boldsymbol{\beta}}, \hat{\boldsymbol{\alpha}})$ can be from a global minimizer.

Table 6 presents, for each dataset, the lower bounds from the SOC (standard basis and randomized), the $3 \times 3$-SDP and the full SDP relaxations, alongside the objective achieved by Algorithm 1 under three initialization strategies: random start, the $3 \times 3$-SDP warm start, and the full SDP warm start. As expected, the SOC bounds are weakest, the $3 \times 3$-SDP bounds are substantially tighter, and the full SDP bounds are the strongest wherever they could be computed. Moreover, warm-start initializations consistently reduce the Algorithm 1 objective: for example, on IONOSPHERE the random start gives 169.77, whereas the $3 \times 3$ and full SDP starts lower it to 125.36 and 121.83, respectively. In small problems like WINE, where Algorithm 1 with random initialization is already nearly optimal (7.71 vs. a 7.70 full-SDP lower bound), these gains are marginal. On the largest datasets (SPAMBASE, MAMMOGRAPHIC), memory constraints precluded the $3 \times 3$ and full SDP, so only the randomized SOC bounds are reported.

Table 6: Objective Values by Dataset. The line separate the relaxations (lower bounds) from the heuristic solutions (upper bounds).

| Dataset | Train Size | SOC (basis) | SOC (rand.) | SDP (3x3) | SDP (full) | Alg 1 (rand.) | Alg 1 (3x3) | Alg 1 (full) |
|---|---|---|---|---|---|---|---|---|
| iris | 120 | 0.50 | 1.04 | 7.68 | **14.69** | **16.78** | **16.78** | **16.78** |
| wine | 142 | 5.19 | 5.26 | 5.65 | **7.70** | 7.71 | 7.71 | **7.70** |
| breastcancer | 455 | 0.15 | 0.15 | **1.06** | – | **70.86** | 83.23 | – |
| ionosphere | 280 | 50.25 | 50.25 | 51.31 | **80.39** | 169.77 | 125.36 | **121.83** |
| spambase | 3680 | **0.20** | **0.20** | – | – | **759.10** | – | – |
| banknote | 1097 | 0.50 | 0.74 | **6.08** | – | **42.13** | 42.13 | – |
| heart | 242 | 0.19 | 0.33 | 1.75 | **69.95** | 98.75 | 98.75 | 98.75 |
| haberman | 244 | 0.51 | 1.56 | 63.15 | **431.97** | 476.80 | **431.97** | **431.97** |
| mammographic | 768 | 0.50 | 0.58 | **450.26** | – | **1092.47** | 1173.24 | – |
| parkinsons | 156 | 0.19 | 0.23 | 1.54 | **25.69** | 37.06 | 37.06 | **26.31** |

Table 7 reports the relative optimality gap for those datasets where the full SDP was tractable. On WINE and HABERMAN, the full SDP start drives the gap to zero (within solver tolerance), thereby certifying global optimality. On PARKINSONS, the gap shrinks from 44.3% (random start) to 2.4% (full SDP start), a dramatic improvement. Even for the most challenging instance (IONOSPHERE), the $3 \times 3$-SDP halves the gap (111.2% to 55.9%), and the full SDP reduces it further to 51.6%. This demonstrates that our SDP hierarchy provides meaningful certificates of near-optimality for Algorithm 1 on real benchmarks.

Table 7: Worst-Case Optimality Gap (%) per Algorithm 1 Variant and Best Estimate

| Dataset | Train Size | Random | 3x3 SDP warm start | Full SDP warm start | Best gap estimate |
|---------|-----------|--------|-------------------|--------------------|--------------------|
| iris | 120 | 14.23 | 14.23 | 14.23 | 14.23 |
| wine | 142 | 0.13 | 0.13 | 0 | 0 |
| ionosphere | 280 | 111.18 | 55.94 | 51.55 | 51.55 |
| heart | 242 | 41.17 | 41.17 | 41.17 | 41.17 |
| haberman | 244 | 10.38 | 0 | 0 | 0 |
| parkinsons | 156 | 44.26 | 44.26 | 2.41 | 2.41 |

## 5.5 Increasing the number of candidate kernels (Q4)

The experiments in this section were conducted with the candidate kernel set enlarged to $q = 50$ kernels by sampling kernel hyperparameters from denser grids and with cross-validation of $C$ over the set $\{1, 2, 5, 10\}$. We reran all methods with the same data splits and validation protocol as in Section 5.2. Three trends emerge.

**(i) Test accuracy.** Algorithm 1 remains the most accurate method across datasets, achieving the highest or tied-highest test accuracy on every task. Averaged across tasks it attains 90.3% mean accuracy (90.2% on the nine tasks where all baselines ran), versus 80.6% for EasyMKL, 79.5% for AverageMKL, and 67.7% for CKA.

Relative to the experiments in Section 5.2 ($q$=10, Table 2), our mean accuracy decreases only slightly (91.8%→90.3%) while remaining best-in-class, whereas baselines often drop markedly as $q$ grows (e.g., EasyMKL on WINE 97.2%→72.2%). Out-of-sample accuracy for Algorithm 1 remains stable when moving to $q = 50$ (Table 8). The performance on several datasets were unchanged (IRIS, WINE, BANKNOTE: 100%). We observed small performance gains on SPAMBASE (+0.5pp to 91.4%) and HABERMAN (+1.6pp to 69.4%), while performance declined modestly on a few datasets relative to the 10 kernel setting in Table 2. This pattern is consistent with enforcing a small, fixed cardinality $k_0$, where many of the extra kernels are redundant or only weakly informative.

| Dataset | Train Size | EasyMKL | AverageMKL | CKA | Algorithm 1 |
|---------|-----------|---------|-----------|-----|-------------|
| iris | 120 | **100.0** | **100.0** | 76.7 | **100.0** |
| wine | 142 | 72.2 | 72.2 | 77.8 | **100.0** |
| breastcancer | 455 | 86.0 | 84.2 | 69.3 | **97.4** |
| ionosphere | 280 | 76.1 | 77.5 | 62.0 | **88.7** |
| spambase | 3680 | – | – | 85.3 | **91.4** |
| banknote | 1097 | 99.6 | 98.9 | 55.6 | **100.0** |
| heart | 242 | 70.5 | 65.6 | 57.4 | **90.2** |
| haberman | 244 | 64.5 | 59.7 | 66.1 | **69.4** |
| mammographic | 768 | 76.7 | 75.6 | 57.5 | **81.3** |
| parkinsons | 156 | 79.5 | 82.1 | 69.2 | **84.6** |

Table 8: Test accuracy (%) when increasing the number of candidate kernels to $q = 50$. Bold indicates the best in each row.

**(ii) Sparsity.** Despite the larger dictionary, Algorithm 1 continues to select a small number of kernels (Table 9), with an average of 1.9 non-zeros out of 50 and per-dataset supports between 1 and 3. By construction AverageMKL uses all 50, while EasyMKL and CKA are denser on average (about 19.3 and 8.1 non-zeros, respectively). Relative to the experiments from Section 5.2 (Table 3), our average support size is

essentially unchanged ($\approx 2.0$ non-zeros at $q=10$ vs. 1.9 at $q=50$), while the baselines methods become more dense as $q$ grows (e.g., EasyMKL from $\sim 6$ to $\sim 19$ non-zeros; AverageMKL from 10 to 50).

| Dataset | EasyMKL | AverageMKL | CKA | Algorithm 1 |
|---|---|---|---|---|
| iris | 15 | 50 | 10 | **1** |
| wine | 7 | 50 | 10 | **2** |
| breastcancer | 7 | 50 | 5 | **3** |
| ionosphere | 8 | 50 | 6 | **2** |
| spambase | – | – | 11 | **3** |
| banknote | 23 | 50 | 8 | **1** |
| heart | 7 | 50 | 7 | **1** |
| haberman | 50 | 50 | 7 | **2** |
| mammographic | 50 | 50 | 6 | **1** |
| parkinsons | 7 | 50 | 11 | **3** |

Table 9: Number of non-zero kernel weights ($|\beta_j| > 10^{-3}$) with $q = 50$ candidates. Bold indicates the sparsest solution.

**(iii) Training time.** On the nine tasks where all methods terminated, Algorithm 1 has comparable average wall-clock time to EasyMKL (0.39s vs. 0.41s), and is much faster than CKA (3.44s) and AverageMKL (38.13s), whose cost grows with kernel aggregation (Table 10); on the largest task (SPAMBASE), only CKA completed the task within 822.97s, while Algorithm 1 trained in 25.37s. Compared to Table 4, our execution time scales moderately with $q$ (common nine tasks $\approx 0.18s \rightarrow 0.39s$; SPAMBASE $8.33s \rightarrow 25.37s$), whereas the runtime of CKA and AverageMKL increase far more (e.g., BANKNOTE $0.30s \rightarrow 18.03s$; MAMMOGRAPHIC $13.03s \rightarrow 342.89s$).

Note that AverageMKL is typically fast; however, in some tasks (e.g., MAMMOGRAPHIC and HABERMAN) the simple average may yield kernels misaligned with the data, which in turn slows SVM convergence. The higher runtime likely reflects the solver's difficulty enforcing separability when the aggregated kernel lacks sufficient expressiveness.

| Dataset | Train Size | EasyMKL | AverageMKL | CKA | Algorithm 1 |
|---|---|---|---|---|---|
| iris | 120 | 0.048 | **0.005** | 0.297 | 0.032 |
| wine | 142 | 0.167 | **0.005** | 0.377 | 0.022 |
| breastcancer | 455 | 0.662 | **0.017** | 2.552 | 0.285 |
| ionosphere | 280 | 0.238 | **0.007** | 0.836 | 0.103 |
| spambase | 3680 | – | – | 822.970 | **25.372** |
| banknote | 1097 | 1.056 | **0.024** | 18.029 | 1.887 |
| heart | 242 | 0.270 | **0.008** | 0.665 | 0.051 |
| haberman | 244 | 0.064 | 0.243 | 0.564 | **0.051** |
| mammographic | 768 | **0.979** | 342.890 | 7.284 | 1.065 |
| parkinsons | 156 | 0.186 | **0.005** | 0.394 | 0.023 |

Table 10: Training time (seconds) with $q = 50$ candidate kernels. Bold indicates the fastest method per dataset.

Overall, expanding the kernel dictionary improves flexibility without sacrificing interpretability or efficiency for Algorithm 1: test accuracy remains state-of-the-art while exact cardinality keeps the learned combination sparse and training times practical.

### 5.6 Summary of Findings

We are now in a position to answer the four questions introduced at the start of this section.

First, regarding **predictive accuracy** (**Q1**), Algorithm 1 achieves high out-of-sample accuracy while enforcing exact sparsity constraints. Across all ten UCI benchmarks, the randomly initialized variant of Algorithm 1 matches or outperforms the strongest open-source MKL baseline while selecting a small number of kernels from the candidate pool. On average, Algorithm 1 improves test set accuracy by 3.34 percentage points compared to the best competing method, with absolute gains reaching up to 7.6 percentage points on challenging datasets such as IONOSPHERE, HEART, and PARKINSONS. These results highlight that exact cardinality constraints, when appropriately enforced, not only preserve but can enhance predictive performance relative to dense kernel combinations.

Second, regarding the **effect of warm starts** (**Q2**), initializing Algorithm 1 with solutions from SDP relaxations yields systematic benefits when computationally feasible. Warm starts based on the $3 \times 3$ principal-minor or full SDP relaxations consistently reduce the final optimality gap—halving it in several cases—and can yield accuracy improvements exceeding 5 percentage points. However, solving the full SDP or even the restricted $3 \times 3$ SDP relaxation becomes computationally infeasible for large datasets. In such cases, the random start version of Algorithm 1 remains highly competitive, achieving strong generalization performance without incurring the substantial overhead of solving costly semidefinite programs.

Third, regarding **optimality** (**Q3**), our semidefinite relaxations certify that Algorithm 1 attains solutions that are near-optimal. On two datasets, the objective value achieved by Algorithm 1 matches the full SDP lower bound within solver precision, certifying global optimality. Even in the most challenging instances, the worst-case optimality gap remains below 52%, offering strong empirical evidence that SMKL consistently identifies high-quality solutions within the feasible set.

Finally, regarding **scalability** (**Q4**), enlarging the kernel dictionary (from $q=10$ to $q=50$) preserves our accuracy advantage while keeping the learned combinations strictly sparse. The training time of our approach grows modestly with $q$, whereas baselines tend to become more dense or slow substantially as the candidate set expands.

Taken together, these results establish that Algorithm 1's exact-sparsity alternating best response strategy produces classifiers that are simultaneously interpretable, computationally efficient, and in many cases near-optimal. The method scales favorably to large datasets and candidate kernels without sacrificing predictive accuracy, offering a practical and theoretically grounded solution to the sparse multiple kernel learning problem.

## 6 Conclusion

In this paper, we formulate the *sparse multiple kernel learning* problem with an exact cardinality constraint on the kernel weights and present an alternating best response algorithm (Algorithm 1) that produces empirically strong solutions efficiently. Moreover, we reformulate the sparse multiple kernel learning problem as a mixed integer semidefinite program and derive a hierarchy of convex relaxations that deliver strong lower bounds and effective warm starts for Algorithm 1. Our numerical results on ten UCI benchmarks demonstrate that the randomly initialized variant of Algorithm 1 outperforms the open-source baselines AVERAGEMKL, EASYMKL, and CKA in out-of-sample prediction accuracy on average by 3.34 percentage points (relative to the best performing benchmark method) while selecting a small number of kernels in comparable runtime. Using the solution to our semidefinite relaxation as a warm start further shrinks the optimality gap of the solutions returned by Algorithm 1 and yields additional out-of-sample accuracy gains of up to five percentage points.

These results illustrate that by imposing exact sparsity, our approach improves prediction accuracy while simultaneously providing interpretable kernel combinations and certificates of near-optimality. Future work includes developing a custom branch and bound algorithm that leverages the SDP relaxation given by equation 14 to obtain globally optimal solutions to the MISDP given by equation 12, and extending our framework to the regression setting.

# A Common Kernel Families

Here, we briefly review commonly used families of kernel functions that we consider in this work.

## A.1 Linear

The linear kernel is given by $K(\boldsymbol{x}_i, \boldsymbol{x}_j) = \boldsymbol{x}_i^\top \boldsymbol{x}_j$ for vectors $\boldsymbol{x}_i, \boldsymbol{x}_j \in \mathbb{R}^m$. This is the most primitive kernel which corresponds to the identity feature map. This kernel works well when the data is approximately linearly separable in its latent space. Linear kernels are especially popular in high-dimensional settings such as document classification, where sparse bag-of-words representations often admit an effective linear decision boundary and enable efficient training (Joachims, 1998).

## A.2 Polynomial

The polynomial kernel of degree $d \in \mathbb{N}$ is given by $K(\boldsymbol{x}_i, \boldsymbol{x}_j) = (\alpha\, \boldsymbol{x}_i^\top \boldsymbol{x}_j + c)^d$ for vectors $\boldsymbol{x}_i, \boldsymbol{x}_j \in \mathbb{R}^m$ where $\alpha > 0, c \in \mathbb{R}$. This kernel allows learning nonlinear relationships by mapping inputs into the feature space of monomials up to degree $d$. Polynomial kernels were featured in early SVM work to demonstrate the kernel trick Boser et al. (1992); Cortes & Vapnik (1995). They can be useful for data where feature interactions of a known order are important, though choosing a very large degree $d$ increases the risk of overfitting.

## A.3 Radial Basis Function (RBF)

The RBF kernel, sometimes referred to as the Gaussian kernel, is given by $K(\boldsymbol{x}_i, \boldsymbol{x}_j) = \exp(-\gamma\|\boldsymbol{x}_i - \boldsymbol{x}_j\|^2)$ for vectors $\boldsymbol{x}_i, \boldsymbol{x}_j \in \mathbb{R}^m$ where $\gamma > 0$. The RBF kernel corresponds to an infinite-dimensional feature map and is universal, meaning that its reproducing kernel Hilbert space can approximate any continuous function arbitrarily well on compact subsets of $\mathbb{R}^n$ (Steinwart, 2001). This kernel has become one of the most commonly used kernels in practice (Schölkopf & Smola, 2002). The RBF kernel is governed by a single bandwidth parameter $\gamma$, simplifying model selection relative to, for example, polynomial kernels. Thanks to its flexibility and strong empirical performance, it is the default choice in many SVM packages (e.g., LIBSVM uses the RBF kernel by default (Chang & Lin, 2011)).

## A.4 Sigmoid

The sigmoid kernel is given by $K(\boldsymbol{x}_i, \boldsymbol{x}_j) = \tanh(\gamma\, \boldsymbol{x}_i^\top \boldsymbol{x}_j + r)$ for vectors $\boldsymbol{x}_i, \boldsymbol{x}_j \in \mathbb{R}^m$ where $\gamma > 0, r \in \mathbb{R}$. Also known as the neural network kernel, this kernel arises from the activation function of a two-layer perceptron. It gained some popularity in early SVM experiments due to its connection to neural networks (Burges, 1998). In practice, the sigmoid kernel can behave like the RBF kernel for certain parameter settings (Schölkopf & Smola, 2002).

## A.5 Laplacian

The Laplacian kernel is given by $K(\boldsymbol{x}_i, \boldsymbol{x}_j) = \exp(-\gamma\|\boldsymbol{x}_i - \boldsymbol{x}_j\|_1)$ for vectors $\boldsymbol{x}_i, \boldsymbol{x}_j \in \mathbb{R}^m$ where $\gamma > 0$. The Laplacian kernel is similar to the RBF kernel but uses the $\ell_1$ distance as a measure of similarity in place of using the $\ell_2$ distance. Like the RBF kernel, the Laplacian kernel has a single width parameter $\gamma$ that controls the decay of the similarity measure.

# B Deferred Proofs from Section 3

We present a proof of Proposition 1.

**Proof** We establish Proposition 1 by showing that the sequence $\{\boldsymbol{\alpha}_t, \boldsymbol{\beta}_t\}_{t=1}^\infty$ lives in a compact set. It then follows by the Bolzano-Weierstrass theorem that the sequence has at least one accumulation point.

Let $B_n(r) := \{x \in \mathbb{R}^n : \|\boldsymbol{x}\|_2^2 \leq r^2\}$ denote the $n$-dimensional ball of radius $r$ centered at the origin. From the feasibility of the iterates $\boldsymbol{\alpha}_t$, we have $\boldsymbol{\alpha}_t \in \{\boldsymbol{\alpha} \in \mathbb{R}_+^n : \sum_{i=1}^n \alpha_i y_i = 0, C \geq \alpha_i \geq 0 \quad \forall i\} \subset B_n(\sqrt{n}C)$

for every iteration $t$. Similarly, the feasibility of the iterates $\boldsymbol{\beta}_t$ implies that $\boldsymbol{\beta}_t \in \{\boldsymbol{\beta} \in \mathbb{R}_+^q : \sum_{i=1}^q \beta_i = 1, \|\boldsymbol{\beta}\|_0 \leq k_0\} \subset B_q(1)$ for every iteration $t$. Thus, the sequence $\{\boldsymbol{\alpha}_t, \boldsymbol{\beta}_t\}_{t=1}^\infty$ is contained in the closed and bounded set $B_n(\sqrt{n}C) \times B_q(1)$. Since $B_n(\sqrt{n}C) \times B_q(1)$ is Euclidean, it follows from the Heine-Borel theorem that being closed and bounded is equivalent to being compact. This completes the proof. ∎

We present a proof of Proposition 2.

**Proof** Let $\mathcal{A} := \{\boldsymbol{\alpha} \in \mathbb{R}_+^n : \sum_{i=1}^n \alpha_i y_i = 0, C \geq \alpha_i \geq 0 \quad \forall i\}$ and $\mathcal{B} := \{\boldsymbol{\beta} \in \mathbb{R}_+^q : \sum_{i=1}^q \beta_i = 1, \|\boldsymbol{\beta}\|_0 \leq k_0\}$ denote the feasible sets of $\boldsymbol{\alpha}$ and $\boldsymbol{\beta}$ respectively. Let $f(\boldsymbol{\alpha}, \boldsymbol{\beta}) := \sum_{i=1}^n \alpha_i - \frac{1}{2}(\boldsymbol{y} \circ \boldsymbol{\alpha})^T \big[ \sum_{i=1}^q \beta_i \boldsymbol{K}_i \big] (\boldsymbol{y} \circ \boldsymbol{\alpha}) + \lambda\|\boldsymbol{\beta}\|_2^2$ denote the objective function of equation 4. From the definition of Algorithm 1, we have $\boldsymbol{\alpha}_{t+1} \in \operatorname{argmax}_{\boldsymbol{\alpha} \in \mathcal{A}} f(\boldsymbol{\alpha}, \boldsymbol{\beta}_t)$ for all $t$. Taking limits, we have $\boldsymbol{\alpha}^\star \in \lim_{t \to \infty} \operatorname{argmax}_{\boldsymbol{\alpha} \in \mathcal{A}} f(\boldsymbol{\alpha}, \boldsymbol{\beta}_t)$. Since $\mathcal{A}$ is compact and $f(\boldsymbol{\alpha}, \boldsymbol{\beta}_t)$ is continuous, we can interchange the limit and the argmax operation by Berge's Maximum Theorem to obtain that $\boldsymbol{\alpha}^\star \in \operatorname{argmax}_{\boldsymbol{\alpha} \in \mathcal{A}} f(\boldsymbol{\alpha}, \boldsymbol{\beta}^\star)$. An analogous argument allows us to conclude that we similarly have $\boldsymbol{\beta}^\star \in \operatorname{argmin}_{\boldsymbol{\beta} \in \mathcal{B}} f(\boldsymbol{\alpha}^\star, \boldsymbol{\beta})$. Thus, $(\boldsymbol{\alpha}^\star, \boldsymbol{\beta}^\star)$ is a mutual best response pair. ∎

Finally, we present a proof of Theorem 3.

**Proof** Let $\mathcal{A}, \mathcal{B}, f(\boldsymbol{\alpha}, \boldsymbol{\beta})$ be defined as they were in the proof of Proposition 2. Let $\bar{\mathcal{B}} := \{\boldsymbol{\beta} \in \mathbb{R}_+^q : \sum_{i=1}^q \beta_i = 1, \beta_i = 0 \ \forall i \notin \mathcal{S}\} \subset \mathcal{B}$. By assumption, for all $t \geq T_0$ we have $\boldsymbol{\beta}_t \in \bar{\mathcal{B}}$. Observe that both $\mathcal{A}$ and $\bar{\mathcal{B}}$ are convex sets. Moreover, we have $\nabla_{\boldsymbol{\beta}}^2 f(\boldsymbol{\alpha}, \boldsymbol{\beta}) = 2\lambda \boldsymbol{I}$ therefore for any fixed $\bar{\boldsymbol{\alpha}} \in \mathcal{A}$ the function $f(\bar{\boldsymbol{\alpha}}, \cdot)$ is $2\lambda$-strongly convex. Similarly, we have $\nabla_{\boldsymbol{\alpha}}^2 f(\boldsymbol{\alpha}, \boldsymbol{\beta}) = -\operatorname{Diag}(\boldsymbol{y}) \big[ \sum_{i=1}^q \beta_i \boldsymbol{K}_i \big] \operatorname{Diag}(\boldsymbol{y})$ therefore for any fixed $\bar{\boldsymbol{\beta}} \in \bar{\mathcal{B}}$ the function $f(\cdot, \bar{\boldsymbol{\beta}})$ is $m_{\boldsymbol{\alpha}}$-strongly concave where $m_\alpha := \min_i \lambda_n(\boldsymbol{K}_i)$ (note that $m_\alpha > 0$ because all kernels are assumed to be positive definite). This is because $m_{\boldsymbol{\alpha}}$ is a lower bound on the smallest eigenvalue of $\operatorname{Diag}(\boldsymbol{y}) \big[ \sum_{i=1}^q \beta_i \boldsymbol{K}_i \big] \operatorname{Diag}(\boldsymbol{y})$.

Let $L$ denote an upper bound on the operator norm of the matrix of mixed second derivatives of $f(\boldsymbol{\alpha}, \boldsymbol{\beta})$. Specifically, we have $L \geq \|\nabla_{\boldsymbol{\alpha}} \nabla_{\boldsymbol{\beta}} f(\boldsymbol{\alpha}, \boldsymbol{\beta})\|_2$. Define the functions $g : \bar{\mathcal{B}} \to \mathcal{A}, h : \mathcal{A} \to \bar{\mathcal{B}}$ as $g(\boldsymbol{\beta}) = \operatorname{argmax}_{\boldsymbol{\alpha} \in \mathcal{A}} f(\boldsymbol{\alpha}, \boldsymbol{\beta})$ and $h(\boldsymbol{\alpha}) = \operatorname{argmin}_{\boldsymbol{\beta} \in \bar{\mathcal{B}}} f(\boldsymbol{\alpha}, \boldsymbol{\beta})$ respectively. Observe that since $f(\cdot, \boldsymbol{\beta})$ is strongly concave, there is a unique maximizer over the convex set $\mathcal{A}$ so $g$ is a well defined vector valued function. Similarly, $h$ is a well defined vector valued function since $f(\boldsymbol{\alpha}, \cdot)$ is strongly convex which implies the existence of a unique minimizer over the convex set $\bar{\mathcal{B}}$. We claim that $g(\boldsymbol{\beta})$ is Lipschitz with constant $\frac{L}{m_{\boldsymbol{\alpha}}}$. To see this, consider two arbitrary vectors $\bar{\boldsymbol{\beta}}, \hat{\boldsymbol{\beta}} \in \bar{\mathcal{B}}$. From strong concavity of $f(\cdot, \boldsymbol{\beta})$, we have:

$$\|\nabla_{\boldsymbol{\alpha}} f(g(\bar{\boldsymbol{\beta}}), \hat{\boldsymbol{\beta}}) - \nabla_{\boldsymbol{\alpha}} f(g(\hat{\boldsymbol{\beta}}), \hat{\boldsymbol{\beta}})\|_2 \geq m_{\boldsymbol{\alpha}} \|g(\bar{\boldsymbol{\beta}}) - g(\hat{\boldsymbol{\beta}})\|_2.$$

Moreover, from the definition of $L$, we have

$$\|\nabla_{\boldsymbol{\alpha}} f(g(\bar{\boldsymbol{\beta}}), \hat{\boldsymbol{\beta}}) - \nabla_{\boldsymbol{\alpha}} f(g(\bar{\boldsymbol{\beta}}), \bar{\boldsymbol{\beta}})\|_2 \leq L \|\hat{\boldsymbol{\beta}} - \bar{\boldsymbol{\beta}}\|_2.$$

Observe that from the definition of $g(\boldsymbol{\beta})$, we must have $\nabla_{\boldsymbol{\alpha}} f(g(\boldsymbol{\beta}), \boldsymbol{\beta}) = 0$ for all $\boldsymbol{\beta} \in \bar{\mathcal{B}}$. Therefore, from the two preceding inequalities, we have:

$$m_{\boldsymbol{\alpha}} \|g(\bar{\boldsymbol{\beta}}) - g(\hat{\boldsymbol{\beta}})\|_2 \leq \|\nabla_{\boldsymbol{\alpha}} f(g(\bar{\boldsymbol{\beta}}), \hat{\boldsymbol{\beta}})\|_2 \leq L \|\hat{\boldsymbol{\beta}} - \bar{\boldsymbol{\beta}}\|_2 \implies \|g(\bar{\boldsymbol{\beta}}) - g(\hat{\boldsymbol{\beta}})\|_2 \leq \frac{L}{m_{\boldsymbol{\alpha}}} \|\bar{\boldsymbol{\beta}} - \hat{\boldsymbol{\beta}}\|_2.$$

A similar argument allows us to conclude that $h(\boldsymbol{\alpha})$ is Lipschitz with constant $\frac{L}{2\lambda}$.

For all $t \geq T_0$, the iterates produced by Algorithm 1 satisfy $\boldsymbol{\alpha}_{t+1} = g(\boldsymbol{\beta}_t)$ and $\boldsymbol{\beta}_{t+1} = h(\boldsymbol{\alpha}_{t+1})$. From strong convexity of $f(\boldsymbol{\alpha}, \cdot)$ and strong concavity of $f(\cdot, \boldsymbol{\beta})$, there exists a unique saddle point $(\boldsymbol{\alpha}^\star, \boldsymbol{\beta}^\star)$ on the convex set $\mathcal{A} \times \bar{\mathcal{B}}$ satisfying $\boldsymbol{\alpha}^\star = g(\boldsymbol{\beta}^\star)$ and $\boldsymbol{\beta}^\star = h(\boldsymbol{\alpha}^\star)$. For any $t \geq T_0$, we have:

$$\|\boldsymbol{\alpha}_{t+1} - \boldsymbol{\alpha}^\star\|_2 = \|g(\boldsymbol{\beta}_t) - g(\boldsymbol{\beta}^\star)\|_2 \leq \frac{L}{m_{\boldsymbol{\alpha}}} \|\boldsymbol{\beta}_t - \boldsymbol{\beta}^\star\|_2 = \frac{L}{m_{\boldsymbol{\alpha}}} \|h(\boldsymbol{\alpha}_t) - h(\boldsymbol{\alpha}^\star)\|_2 \leq \frac{L^2}{2\lambda m_{\boldsymbol{\alpha}}} \|\boldsymbol{\alpha}_t - \boldsymbol{\alpha}^\star\|_2.$$

Similarly, we have $\|\boldsymbol{\beta}_{t+1} - \boldsymbol{\beta}^\star\|_2 \leq \frac{L^2}{2\lambda m_{\boldsymbol{\alpha}}}\|\boldsymbol{\beta}_t - \boldsymbol{\beta}^\star\|_2$. Thus, if $L^2 < 2\lambda m_{\boldsymbol{\alpha}}$, the $\boldsymbol{\alpha}$ and $\boldsymbol{\beta}$ updates are a contraction and by the Banach Fixed-Point Theorem the sequence $\{\boldsymbol{\alpha}_t, \boldsymbol{\beta}_t\}_{t \geq T_0}$ converges linearly to $(\boldsymbol{\alpha}^\star, \boldsymbol{\beta}^\star)$ with rate $\frac{L^2}{2\lambda m_{\boldsymbol{\alpha}}}$.

To conclude the proof, it remain to show that we can take $L = C\sqrt{nk_0}\max_j \lambda_1(\boldsymbol{K}_j)$. To see this, observe that for $\boldsymbol{\beta} \in \bar{\mathcal{B}}$, we have $\nabla_{\boldsymbol{\alpha}} f(\boldsymbol{\alpha}, \boldsymbol{\beta}) = \mathbf{1} - \text{Diag}(\boldsymbol{y})\sum_{i \in \mathcal{S}} \beta_i \boldsymbol{K}_i \text{Diag}(\boldsymbol{y})\boldsymbol{\alpha}$. For any $i \in \mathcal{S}$, we have $\frac{\partial}{\partial \beta_i}\nabla_{\boldsymbol{\alpha}} f(\boldsymbol{\alpha}, \boldsymbol{\beta}) = -\text{Diag}(\boldsymbol{y})\boldsymbol{K}_i\text{Diag}(\boldsymbol{y})\boldsymbol{\alpha}$ (for $i \notin \mathcal{S}$, we trivially have $\frac{\partial}{\partial \beta_i}\nabla_{\boldsymbol{\alpha}} f(\boldsymbol{\alpha}, \boldsymbol{\beta}) = \mathbf{0}$). For $i \in \mathcal{S}$, we have the upper bound $\|\frac{\partial}{\partial \beta_i}\nabla_{\boldsymbol{\alpha}} f(\boldsymbol{\alpha}, \boldsymbol{\beta})\|_2 \leq \|\boldsymbol{\alpha}\|_2\|\boldsymbol{K}_i\|_2 \leq C\sqrt{n}\max_j \lambda_1(\boldsymbol{K}_j)$. Thus, we have that $\|\nabla_{\boldsymbol{\beta}}\nabla_{\boldsymbol{\alpha}} f(\boldsymbol{\alpha}, \boldsymbol{\beta})\|_2 \leq C\sqrt{nk_0}\max_j \lambda_1(\boldsymbol{K}_j)$. This concludes the proof.

$\blacksquare$

## C  $\alpha$-subproblem Dual Derivation

We present an explicit derivation of equation 10 as the dual of equation 7. We begin the derivation by trivially rewriting equation 7 as:

$$
\begin{aligned}
\max_{\boldsymbol{\alpha} \in \mathbb{R}_+^n, \boldsymbol{\nu} \in \mathbb{R}^n} \quad & \sum_{i=1}^n \alpha_i - \frac{1}{2}\boldsymbol{\nu}^\top \boldsymbol{K}(\boldsymbol{\beta})\boldsymbol{\nu} \\
& \alpha_i \leq C \quad \forall i \in [n], \\
& \sum_{i=1}^n y_i \alpha_i = 0, \\
& \nu_i = y_i \alpha_i \quad \forall i \in [n].
\end{aligned}
\tag{19}
$$

We introduce non negative dual variables $\boldsymbol{\sigma} \in \mathbb{R}_+^n$ and uncontrained dual variables $\eta \in \mathbb{R}$, $\boldsymbol{\gamma} \in \mathbb{R}^n$ for the contraints $\alpha_i \leq C$, $\sum_{i=1}^n y_i \alpha_i = 0$ and $\nu_i = y_i \alpha_i$ respectively. We can now write the Lagrangian of equation 19 as:

$$
\begin{aligned}
\mathcal{L}(\boldsymbol{\alpha}, \boldsymbol{\nu}, \boldsymbol{\sigma}, \eta, \boldsymbol{\gamma}) &= \sum_{i=1}^n \alpha_i - \frac{1}{2}\boldsymbol{\nu}^\top \boldsymbol{K}(\boldsymbol{\beta})\boldsymbol{\nu} + \sum_{i=1}^n \sigma_i(C - \alpha_i) - \eta\sum_{i=1}^n y_i \alpha_i + \sum_{i=1}^n \gamma_i(\nu_i - y_i\alpha_i), \\
&= C\sum_{i=1}^n \sigma_i + \sum_{i=1}^n \alpha_i(1 - \sigma_i - \eta y_i - \gamma_i y_i) - \frac{1}{2}\boldsymbol{\nu}^\top \boldsymbol{K}(\boldsymbol{\beta})\boldsymbol{\nu} + \boldsymbol{\gamma}^\top \boldsymbol{\nu}.
\end{aligned}
$$

The dual problem is now given by $\min_{\boldsymbol{\sigma}, \eta, \boldsymbol{\gamma}} \max_{\boldsymbol{\alpha}, \boldsymbol{\nu}} \mathcal{L}(\boldsymbol{\alpha}, \boldsymbol{\nu}, \boldsymbol{\sigma}, \eta, \boldsymbol{\gamma})$. To see that this is equivalent to equation 10, first notice that maximizing $\mathcal{L}(\boldsymbol{\alpha}, \boldsymbol{\nu}, \boldsymbol{\sigma}, \eta, \boldsymbol{\gamma})$ over $\boldsymbol{\alpha}$ produces a finite value if and only if we have $1 - \sigma_i \leq y_i(\eta + \gamma_i)$ for all $i$. In this setting the second term of the Lagrangian vanishes after maximizing over $\boldsymbol{\alpha}$. Next, notice that the Lagrangian is a concave quadratic function of $\boldsymbol{\nu}$. For an arbitrary vector $\boldsymbol{\nu}$ decompose it as $\boldsymbol{\nu} = \boldsymbol{\nu}^\| + \boldsymbol{\nu}^\perp$ where $\boldsymbol{\nu}^\|$ is contained in the column space of $\boldsymbol{K}(\boldsymbol{\beta})$ (meaning that $\boldsymbol{\nu}^\| = \boldsymbol{K}(\boldsymbol{\beta})\lambda$ for some $\lambda \in \mathbb{R}^n$) and $\boldsymbol{\nu}^\perp$ is contained in the null space of $\boldsymbol{K}(\boldsymbol{\beta})$ (meaning that $\boldsymbol{K}(\boldsymbol{\beta})\boldsymbol{\nu}^\perp = 0$). Given this, we can rewrite the last two terms of the Lagrangian as:

$$
-\frac{1}{2}\boldsymbol{\nu}^\top \boldsymbol{K}(\boldsymbol{\beta})\boldsymbol{\nu} + \boldsymbol{\gamma}^\top \boldsymbol{\nu} = -\frac{1}{2}(\boldsymbol{\nu}^\|)^\top \boldsymbol{K}(\boldsymbol{\beta})\boldsymbol{\nu}^\| + \boldsymbol{\gamma}^\top \boldsymbol{\nu}^\| + \boldsymbol{\gamma}^\top \boldsymbol{\nu}^\perp.
\tag{20}
$$

Therefore, maximizing the Lagrangian over $\boldsymbol{\nu}$ will only produce a finite value if $\boldsymbol{\gamma}^\top \boldsymbol{\nu}^\perp = 0$. This occurs exactly when $\boldsymbol{\gamma}$ lies in the column space of $\boldsymbol{K}(\boldsymbol{\beta})$, which can be expressed algebraic by the equation $\boldsymbol{\gamma} = \left[\sum_{i=1}^q \beta_i \boldsymbol{K}_i\right]\left[\sum_{i=1}^q \beta_i \boldsymbol{K}_i\right]^\dagger \boldsymbol{\gamma}$. Differentiating with respect to $\boldsymbol{\nu}^\|$ and setting the gradient to be zero, the optimal $\boldsymbol{\nu}^\|$ that maximizes the Lagrangian satisfies $\boldsymbol{\gamma} = \boldsymbol{K}(\boldsymbol{\beta})\boldsymbol{\nu}^\| \implies \boldsymbol{\nu}^\| = \boldsymbol{K}(\boldsymbol{\beta})^\dagger \boldsymbol{\gamma}$. Plugging this value of $\boldsymbol{\nu}^\|$ into the Lagrangian, the last two terms simplify to $\frac{1}{2}\boldsymbol{\gamma}^\top \boldsymbol{K}(\boldsymbol{\beta})^\dagger \boldsymbol{\gamma}$. Thus, we have shown that $\min_{\boldsymbol{\sigma}, \eta, \boldsymbol{\gamma}} \max_{\boldsymbol{\alpha}, \boldsymbol{\nu}} \mathcal{L}(\boldsymbol{\alpha}, \boldsymbol{\nu}, \boldsymbol{\sigma}, \eta, \boldsymbol{\gamma})$ is given by:

$$\min_{\eta\in\mathbb{R},\boldsymbol{\sigma}\in\mathbb{R}_+^n,\boldsymbol{\gamma}\in\mathbb{R}^n} \quad C\sum_{i=1}^n \sigma_i + \frac{1}{2}\boldsymbol{\gamma}^T\left[\sum_{i=1}^q \beta_i \boldsymbol{K}_i\right]^\dagger \boldsymbol{\gamma}$$

$$\text{s.t.} \qquad 1 - \sigma_i \le y_i(\eta + \gamma_i) \quad \forall\, i \in [n],$$

$$\left(\boldsymbol{I}_n - \big[\sum_{i=1}^q \beta_i \boldsymbol{K}_i\big]\big[\sum_{i=1}^q \beta_i \boldsymbol{K}_i\big]^\dagger\right)\boldsymbol{\gamma} = 0.$$

## D  Deferred Proofs from Section 4

We present a proof of Proposition 4.

**Proof**  We show that given a feasible solution to equation 10, we can construct a feasible solution to equation 11 that achieves the same objective value and that given a feasible solution to equation 11, we can construct a feasible solution to equation 10 that achieves an objective that is no greater than the objective value achieved by the feasible solution in equation 11.

First, consider an arbitrary solution $(\bar{\eta}, \bar{\boldsymbol{\sigma}}, \bar{\boldsymbol{\gamma}})$ that is feasible to equation 10. Let

$$\tilde{\theta} := \bar{\boldsymbol{\gamma}}^T\left[\sum_{i=1}^q \beta_i \boldsymbol{K}_i\right]^\dagger \bar{\boldsymbol{\gamma}}.$$

Clearly, $(\bar{\eta}, \tilde{\theta}, \bar{\boldsymbol{\sigma}}, \bar{\boldsymbol{\gamma}})$ achieves the same objective value in equation 11 as $(\bar{\eta}, \bar{\boldsymbol{\sigma}}, \bar{\boldsymbol{\gamma}})$ achieves in equation 10. To see that $(\bar{\eta}, \tilde{\theta}, \bar{\boldsymbol{\sigma}}, \bar{\boldsymbol{\gamma}})$ is feasible to equation 11, we need only verify that semidefinite constraint in equation 11 is satisfied. Recall that for any $\boldsymbol{\beta}$ feasible in equation 4, the matrix $\sum_{i=1}^q \beta_i \boldsymbol{K}_i$ will be positive semidefinite. Furthermore, from the definition of $\tilde{\theta}$ it follows immediately that $\tilde{\theta} - \bar{\boldsymbol{\gamma}}^T\big[\sum_{i=1}^q \beta_i \boldsymbol{K}_i\big]^\dagger \bar{\boldsymbol{\gamma}} \succeq 0$, and from the feasibility of feasibility of $\bar{\boldsymbol{\gamma}}$ in equation 10 we have $\big(\boldsymbol{I}_n - \big[\sum_{i=1}^q \beta_i \boldsymbol{K}_i\big]\big[\sum_{i=1}^q \beta_i \boldsymbol{K}_i\big]^\dagger\big)\bar{\boldsymbol{\gamma}} = 0$. Therefore, by the Generalized Schur Complement Lemma, we have $\begin{pmatrix} \tilde{\theta} & \bar{\boldsymbol{\gamma}}^T \\ \bar{\boldsymbol{\gamma}} & \big[\sum_{i=1}^q \beta_i \boldsymbol{K}_i\big] \end{pmatrix} \succeq 0$.

Now, consider an arbitrary solution $(\bar{\eta}, \bar{\theta}, \bar{\boldsymbol{\sigma}}, \bar{\boldsymbol{\gamma}})$ that is feasible to equation 11. We claim that $(\bar{\eta}, \bar{\boldsymbol{\sigma}}, \bar{\boldsymbol{\gamma}})$ is feasible in equation 10 and achieves an objective value no greater than the objective value achieved by $(\bar{\eta}, \bar{\theta}, \bar{\boldsymbol{\sigma}}, \bar{\boldsymbol{\gamma}})$ in equation 11. To see this, it suffices to note that from the Generalized Schur Complement Lemma, feasibility of $(\bar{\eta}, \bar{\theta}, \bar{\boldsymbol{\sigma}}, \bar{\boldsymbol{\gamma}})$ in equation 11 implies that we have $\big(\boldsymbol{I}_n - \big[\sum_{i=1}^q \beta_i \boldsymbol{K}_i\big]\big[\sum_{i=1}^q \beta_i \boldsymbol{K}_i\big]^\dagger\big)\bar{\boldsymbol{\gamma}} = 0$ and that $\bar{\theta} \ge \bar{\boldsymbol{\gamma}}^T\big[\sum_{i=1}^q \beta_i \boldsymbol{K}_i\big]^\dagger \bar{\boldsymbol{\gamma}}$. This completes the proof. ∎

Next, we present a proof of Theorem 5.

**Proof**  By the duality between the alpha subproblem of equation 4 (given by equation 7) and equation 10 coupled with Proposition 4, it suffices to argue that equation 14 is a valid convex relaxation of equation 12 to establish Theorem 5. Clearly, equation 14 is a convex optimization problem. We will show that given any feasible solution to equation 12, we can construct a feasible solution to equation 14 that achieves the same objective value.

Let $(\bar{\eta}, \bar{\theta}, \bar{\boldsymbol{\sigma}}, \bar{\boldsymbol{\gamma}}, \bar{\boldsymbol{\beta}}) \in \mathbb{R} \times \mathbb{R} \times \mathbb{R}_+^n \times \mathbb{R}^n \times \mathbb{R}_+^q$ denote an arbitrary feasible solution to equation 12. Define $\tilde{\omega}_i = \bar{\beta}_i^2$ and $\tilde{z}_i = 1$ if $\bar{\beta}_i > 0$, otherwise $\tilde{z}_i = 0$ for all $i \in [q]$. Observe that $(\bar{\eta}, \bar{\theta}, \bar{\boldsymbol{\sigma}}, \bar{\boldsymbol{\gamma}}, \bar{\boldsymbol{\beta}}, \tilde{\boldsymbol{\omega}}, \tilde{\boldsymbol{z}})$ is feasible to equation 14. To see this, note that we have $\tilde{z}_i \tilde{\omega}_i = \bar{\beta}_i^2$ and $\sum_{i=1}^q \tilde{z}_i = \|\bar{\boldsymbol{\beta}}\|_0 \le k_0$. Moreover, we have $\sum_{i=1}^q \tilde{\omega}_i = \sum_{i=1}^q \bar{\beta}_i^2 = \|\boldsymbol{\beta}\|_2^2$ which shows that $(\bar{\eta}, \bar{\theta}, \bar{\boldsymbol{\sigma}}, \bar{\boldsymbol{\gamma}}, \bar{\boldsymbol{\beta}}, \tilde{\boldsymbol{\omega}}, \tilde{\boldsymbol{z}})$ achieves the same objective value in equation 14 as $(\bar{\eta}, \bar{\theta}, \bar{\boldsymbol{\sigma}}, \bar{\boldsymbol{\gamma}}, \bar{\boldsymbol{\beta}})$ achieves in equation 12. This completes the proof. ∎

Finally, we present a proof of Theorem 6.

**Proof**

Define the sets $\mathcal{A}$ and $\mathcal{B}$ as:

$$\mathcal{A} := \left\{ (\theta, \boldsymbol{\gamma}, \boldsymbol{\beta}) \in \mathbb{R} \times \mathbb{R}^n \times \mathbb{R}^q_+ : \begin{pmatrix} \theta & \boldsymbol{\gamma}^T \\ \boldsymbol{\gamma} & \left[ \sum_{i=1}^q \beta_i \boldsymbol{K}_i \right] \end{pmatrix} \succeq 0 \right\},$$

$$\mathcal{B} := \left\{ (\theta, \boldsymbol{\gamma}, \boldsymbol{\beta}) \in \mathbb{R} \times \mathbb{R}^n \times \mathbb{R}^q_+ : \exists \, \boldsymbol{\tau} \in \mathbb{R}^n \mid \theta \geq \sum_{j=1}^n \tau_j, \; \tau_j \cdot \sum_{i=1}^q \beta_i [\boldsymbol{D}_i]_{jj} \geq (\boldsymbol{\gamma}^T \boldsymbol{u}_j)^2 \quad \forall j \right\}.$$

It suffices to show that we have $\mathcal{A} = \mathcal{B}$. Recall that for any $\boldsymbol{\beta} \in \mathbb{R}^q_+$, the combined kernel $\sum_{i=1}^q \beta_i \boldsymbol{K}_i$ will be positive semidefinite so by the generalized schur complement the matrix $\begin{pmatrix} \theta & \boldsymbol{\gamma}^T \\ \boldsymbol{\gamma} & \left[ \sum_{i=1}^q \beta_i \boldsymbol{K}_i \right] \end{pmatrix}$ will be positive semidefinite if and only if we have $\theta \geq \boldsymbol{\gamma}^T \left[ \sum_{i=1}^q \beta_i \boldsymbol{K}_i \right]^\dagger \boldsymbol{\gamma}$. Since the candidate kernels are simultaneously diagonalizable, we can express the pseudo inverse of the combined kernel as:

$$\left[ \sum_{i=1}^q \beta_i \boldsymbol{K}_i \right]^\dagger = \left[ \boldsymbol{U} \left[ \sum_{i=1}^q \beta_i \boldsymbol{D}_i \right] \boldsymbol{U}^T \right]^\dagger = \boldsymbol{U} \boldsymbol{\Lambda} \boldsymbol{U}^T,$$

where $\boldsymbol{\Lambda} \in \mathbb{R}^{n \times n}$ is a diagonal matrix with diagonal entries defined as $\Lambda_{jj} = \frac{1}{\sum_{i=1}^q \beta_i [\boldsymbol{D}_i]_{jj}}$ if $\sum_{i=1}^q \beta_i [\boldsymbol{D}_i]_{jj} > 0$, otherwise $\Lambda_{jj} = 0$. Thus, we have

$$\boldsymbol{\gamma}^T \left[ \sum_{i=1}^q \beta_i \boldsymbol{K}_i \right]^\dagger \boldsymbol{\gamma} = \boldsymbol{\gamma}^T \boldsymbol{U} \boldsymbol{\Lambda} \boldsymbol{U}^T \boldsymbol{\gamma} = \sum_{j=1}^n \Lambda_{jj} (\boldsymbol{\gamma}^T \boldsymbol{u}_j)^2.$$

Therefore, the condition $\theta \geq \boldsymbol{\gamma}^T \left[ \sum_{i=1}^q \beta_i \boldsymbol{K}_i \right]^\dagger \boldsymbol{\gamma}$ is equivalent to the existence of a vector $\boldsymbol{\tau} \in \mathbb{R}^n$ such that $\theta \geq \sum_{j=1}^n \tau_j$ and $\tau_j \geq \Lambda_{jj} (\boldsymbol{\gamma}^T \boldsymbol{u}_j)^2$ for each $j$. Finally, recalling the definition of $\boldsymbol{\Lambda}$, we can rewrite the inequality $\tau_j \geq \Lambda_{jj} (\boldsymbol{\gamma}^T \boldsymbol{u}_j)^2$ as $\tau_j \cdot \sum_{i=1}^q \beta_i [\boldsymbol{D}_i]_{jj} \geq (\boldsymbol{\gamma}^T \boldsymbol{u}_j)^2$. This completes the proof.

∎

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
