# OpenReview forum: "Sparse Multiple Kernel Learning: Alternating Best Response and Semidefinite Relaxations"
_TMLR — Accepted by TMLR_

### Review · Reviewer_PSAX · 2025-08-31

**Summary Of Contributions:**

This paper addresses Sparse Multiple Kernel Learning (SMKL) by directly imposing cardinality constraints on kernel weights rather than using $\ell_1$ regularization as a sparsity surrogate. To solve the non-convex optimization problem, the authors propose an alternating best response algorithm that iterates between solving a standard SVM dual ($\alpha$-subproblem) and a sparse kernel weight selection problem ($\beta$-subproblem). In both cases, with one subproblem fixed, the remaining subproblem becomes convex and tractable. The $\alpha$-subproblem can be solved efficiently using existing solvers like LIBSVM. The exact minimizer of the $\beta$-subproblem can be attained in quasi-linear time.

**Audience:**

Yes

**Audience Explanation:**

The paper presents interesting algorithmic ideas and insightful numerical results.

**Broader Impact Concerns:**

N.A.

**Claims And Evidence:**

No

**Claims Explanation:**

The paper presents comprehensive numerical results demonstrating the superior performance of the proposed method across multiple UCI benchmarks, but I have some concerns regarding the absence of theoretical guarantees that would support these empirical claims.

The authors prove that their mixed integer semidefinite reformulation is equivalent to the original problem. They also show that their SDP relaxation (from $z\in \{0, 1\}^q$ to $z\in [0, 1]^q$) provide a valid lower bound.

However, besides these, **i)** there doesn't seem to be convergence analysis of the algorithm; **ii)** it's hard to argue how close the heuristic solutions are to the global minima since the lower bounds based on SDP relaxations may be quite loose; **iii)** there is no generalization error analysis justifying why we should expect that the exact cardinality constraints improve generalization performance over $\ell_1$ surrogates; **iv)** it would be very useful to include an analysis of the algorithm's computational complexity.

**Requested Changes:**

- Sentences like “We present … that produces high quality solutions to Problem 4…” appear multiple times throughout the paper as a description of the main contribution of the paper, but I wonder what exactly “high quality” means. Apart from empirical results, are there any theoretical guarantees on the improvements in predictive accuracy and runtime that the proposed algorithm can achieve? Please see above for the four types of guarantees i)--iv) which I think could largely strengthen the paper.

- p.15: all numerical results consider 10 base kernels. Might 10 be too small to compare various sparsity-inducing methods, as the advantages of sparse selection become most apparent when choosing from larger candidate sets?

- Table 7: I have a question about the certification of near global optimality here. If I understand correctly, the certification is only meaningful when the SDP relaxation happens to be tight. The tightness of this relaxation isn't proven and is only verified post-hoc by observing small gaps in some cases. Is this correct? For the results for breastcancer, spambase, mammographic etc in Table 7, the reported optimality gap seems to be very large, so large that we probably cannot claim optimality here at all. Am I missing something here?

- Minor: For clarity, consider using \\\eqref{} or Eq.~\\\eqref{} to reference equations instead of “Problem xx”

---

> ### Author Response · Authors · 2025-10-25
> **Thank You Reviewer PSAX - Response**
>
> Thank you for your thoughtful feedback on our manuscript, which has improved the quality of our paper. We have revised the manuscript based on your comments and those of the other referees. We believe we have fully incorporated all of your feedback. We now respond to each of your requested changes below, point by point.
>
>
> Requested change 1:
>
> Thank you for this comment. By “produces high quality solutions to Problem 4...”, we were alluding to the empirical results that demonstrate that the solutions returned by our algorithm outperform the benchmark methods, and that for several of the small scale datasets we are able to exhibit a matching lower bound. To increase clarity in the revised manuscript, we have removed occurrences of the phrase “high quality” in favor of more descriptive language.
>
> Thank you for your very detailed description of the types of theoretical insights that would be relevant for our work. In the newly added Section 3.4 of the revised manuscript, we have added a convergence analysis of our algorithm (Proposition 1, Proposition 2, Theorem 3 - type i) and an analysis of the algorithm’s computational complexity (type iv). Regarding type ii, we certainly agree that on larger datasets the SDP lower bounds can be quite loose. In practice, if we wanted to guarantee obtaining a feasible solution with a matching lower bound, we could embed our heuristic and our SDP relaxation within a custom branch and bound algorithm as described in Section 4.2 (at the expense of runtime). While this would not create a theoretical guarantee, empirically the solution found at the root node in such branch and bound algorithms is nearly optimal and the majority of the runtime of branch and bound is spent improving the lower bound by recursively partitioning the parameter space. See for example [1, 2]. We acknowledge that we do not include a theoretical generalization analysis (type iii), however we think that the newly added theoretical results alongside the existing and expanded empirical evidence sufficiently exhibit the advantages of our algorithm. Thank you again for this comment which we think has helped substantially improve the quality of the manuscript.
>
>
> Requested change 2:
>
> Thank you for the suggestion. We have added a new research question (Q4) that explores what happens when we expand the candidate kernel set to q = 50. In this setting, our method maintains its accuracy advantage while keeping the learned combinations sparse (1–3 active kernels), and training time grows only mildly with q. By contrast, baseline methods tend to become more dense or slow substantially as the dictionary expands. We summarize these results in the revised manuscript. Thank you again for this comment.
>
>
> Requested change 3:
>
> Thank you for this comment. The certifications should be interpreted as worst case guarantees. So, for example, a computed gap of say 15% means that the objective value achieved by the output of our algorithm is no worse than 15% larger than the optimal objective value (but it could in fact be closer). In this regard, the certifications are always meaningful. What you are alluding to is that an exact optimality guarantee (gap of 0%) occurs only when the SDP relaxation happens to be exactly tight which we certainly agree with. When the optimality gap is large, we do not claim the solution is optimal but rather claim a worst case guarantee on how far the returned solution is from the optimal one. You are correct that we do not prove tightness of the relaxation and in general it will not be tight (otherwise we could just solve the relaxation to obtain an optimal feasible solution). The relaxation strength is only verified post-hoc as is typical for non-tight convex relaxations of non-convex problems. As we alluded to in our response to your first comment, to guarantee obtaining a feasible solution with a matching lower bound, one could embed our heuristic algorithm alongside the SDP relaxation within a custom branch and bound algorithm. Thank you again for this comment.
>
>
> Requested change 4:
>
> Thank you for this comment. We have incorporated this suggestion into the revised manuscript.
>
>
> [1] Dimitris Bertsimas and Nicholas AG Johnson. Compressed sensing: A discrete optimization approach.
> Machine Learning, 113(9):6725–6764, 2024.
>
> [2] Dimitris Bertsimas, Ryan Cory-Wright, and Nicholas A. G. Johnson. Sparse plus low rank matrix
> decomposition: A discrete optimization approach. Journal of Machine Learning Research, 24(267):1–51,
> 2023.

---

### Review · Reviewer_Pmss · 2025-09-14

**Summary Of Contributions:**

This paper addresses sparse multiple kernel learning by introducing a novel formulation that imposes an explicit cardinality constraint on kernel weights, combined with an L2 penalty. To be specific, an additional <= constraint of non-zero elements is added to the L1-regularized kernel weights, offering extra control over sparsity.

**Audience:**

Yes

**Audience Explanation:**

Solving MKL problem with explicit cardinality constraint is an important research topic.

**Claims And Evidence:**

Yes

**Claims Explanation:**

1. The alternating best response algorithm is well-designed, leveraging efficient subproblem solvers (LIBSVM for $\alpha$, custom greedy/projection for $\beta$).
2. The derivation of semidefinite convex relaxations provides a theoretical foundation for certifying solution quality.
3. Comprehensive empirical evaluation on multiple UCI datasets demonstrates consistent and substantial improvements over existing MKL methods, with clear ablation (random vs. warm start) and runtime comparisons.

**Requested Changes:**

1. Discuss the feasibility of imposing both >= and <= cardinality constraints.
2. Include computational complexity analyses
3. Considering moving Literature Review and Preliminaries to supplementary material to streamline the main text and focus on novel contributions.

---

> ### Author Response · Authors · 2025-10-25
> **Thank You Reviewer Pmss - Response**
>
> Thank you very much for your review of our manuscript and for your positive evaluation. We now respond to each of your requested changes below, point by point.
>
>
> Requested change 1:
>
> Thank you for this comment. Our approach should be amenable to imposing both upper and lower bounds on the cardinality of $\beta$ through the constraint $k_0 \leq \Vert \beta \Vert_0 \leq k_1$ provided suitable modifications to the GSSP algorithm to exactly solve the $\beta$ subproblem with this additional lower bound constraint. Our newly added theoretical results would be preserved in this setting. That being said, it is unclear what the practical motivation would be to impose such a lower bound since for a given objective value we desire the sparsest kernel. The feasible set of $k_0 \leq \Vert \beta \Vert_0 \leq k_1$ is a strict subset of $ \Vert \beta \Vert_0 \leq k_1$ so the solution of the former case would be no less sparse and no more accurate than the solution of the latter case. Thank you again for this comment.
>
>
> Requested change 2:
>
> Thank you for this comment. In the revised manuscript, we have included an analysis of the computational complexity as well as the theoretical convergence of Algorithm 1 in Section 3.4.
>
>
> Requested change 3:
>
> Thank you for this comment. In the revised manuscript, we have moved the preliminaries to Appendix A.

---

### Review · Reviewer_c5bU · 2025-10-07

**Summary Of Contributions:**

This paper presents new approaches to learn a sparse kernel function from a linear combination of kernel functions. The paper imposes an explicit $\ell_0$ constraint to the basis combination coefficients, and proposes to solve the optimization problem by alternative minimization strategies. The update of the weight vector can be done by standard LIBSVM solvers. The update of combination coefficients can be exactly computed in quasilinear time. The paper further proposes an exact reformation as a mixed integer semidefinite optimization problem and considers the convex relaxation. The paper also presents extensive experimental comparisons with three baseline methods to check the efficiency of the proposed algorithm.

**Audience:**

Yes

**Audience Explanation:**

Learning good kernels are effective to improve the representation power of models since a kernel function corresponds to a feature. MKL provides a principled method to learn the kernel from datasets. This paper proposes a new MKL method which can improve the sparsity. The paper also gives several reformulations of the problem to improve the implementation efficiency. These techniques should be interesting to TMLR community.

**Broader Impact Concerns:**

No issues on ethical implications.

**Claims And Evidence:**

Yes

**Claims Explanation:**

**Strength**

- The paper proposes to learn a sparse combination of kernel functions by imposing a $\ell_0$ constraint. A nice property is that the update of the kernel coefficients can be exactly computed. This is impressive since the $\ell_0$ constraint is highly challenging to handle in general.

- The paper also gives interesting reformulations based on mixed integer semidefinite optimization problems, which is useful to design guidelines to certify optimality of the solution.

- The authors open-resources the code to a website.

- The authors also make an empirical comparison on test accuracy, execution time, sparsity and optimality gap and the initialization strategies. These experimental comparisons show consistent improvements over baseline methods.


**Weakness**
- The proposed methods consider $\ell_0$, $\ell_1$ and $\ell_2$ regularizer/constraints. This introduces two hyperparameters to tune. Furthermore, the proposed method also needs to tune the hyperparameter $C$. This may cause high computation costs to tune these hyperparameters.

- The paper considers three baseline methods EasyMKL, averageMKL and CKA. These seem to be not the most strong baseline methods in MKL. For example, another strong baseline method is $\ell_p$-norm MKL MKL introduced in [1]. It would be interesting to make a comparison with this baseline.



[1] Marius Kloft, Ulf Brefeld, Soeren Sonnenburg, and Alexander Zien. $\ell_p$-norm multiple kernel learning. Journal of Machine Learning Research, 12:953–997, 2011.

**Requested Changes:**

- It would be interesting to consider strong baseline methods such as $\ell_p$-norm MKL in the experimental comparison.

- The structure can be improved. For example, the paper includes some proofs in Section 4. It would be better to collect these proofs in a single section, or put them to appendix.

- In Section 4.1, the paper shows that the dual problem of the alpha subproblem in Problem 4 is Problem 10. It is not quite clear to me why this holds. It would be nice if the authors can give some explanations.

- In table 4, the paper shows that AverageMKL requires 1.9 and 13 seconds for two datasets, which are much larger than other methods. This is a bit surprising since AverageMKL simply uses the average of base kernels as the kernel function. Then, it should achieve the smallest computational cost. It would be nice if the authors can give some explanations.

- Theorem 3 requires that the kernels are simultaneously diagonalizable. This is a strong assumption. In practice, most kernel matrices cannot be simultaneously diagonalizable.

Minor issues
- The authors should be consistent in the use of notations. For example, $m$ and $n$ are used for dimensionality and sample size, respectively in Section 1. However, in Section 2, $n$ is used for dimensionality.
- Section 2.1.2: "a useful" should be "an useful"
- Section 4.1: "It's dual" should be "Its dual"
- Section 4.2.1: "$n+1 \times n+1$" should be "$(n+1)\times(n+1)$"
- Section 5.4: "we can we can" should be "we can"

---

> ### Author Response · Authors · 2025-10-25
> **Thank You Reviewer c5bU - Response**
>
> Thank you for your thoughtful feedback on our manuscript, which has improved the quality of our paper. We have revised the manuscript based on your comments and those of the other referees. We believe we have fully incorporated all of your feedback. We now respond to each of your comments below, point by point.
>
>
> Requested change 1:
>
> Thank you for this comment. We attempted to find a Python or Julia based implementation of the $\ell_p$ method you mentioned but were unable to find one. We did find a MATLAB implementation, however we were unable to find appropriate code documentation so were unable to benchmark against this approach. That being said, in the revised manuscript we have extended our numerical results to illustrate the performance of our algorithm when the candidate kernel set is much larger than in the initial set of experiments. We have additional significantly expanded our theoretical results (Section 3.4) to explore the convergence properties and computational complexity of our algorithm. We think that this expanded body of evidence sufficiently presents exhibits the advantages of our algorithm. Thank you again for this comment.
>
>
> Requested change 2:
>
> Thank you for this suggestion. In the revised manuscript, we have moved all proofs from Section 4 to
> Appendix D.
>
>
> Requested change 3:
>
> Thank you for this comment. In the revised manuscript, we have moved included an explicit derivation of Problem 10 starting from Problem 7 in Appendix C.
>
>
> Requested change 4:
>
> Thank you for the comment. Our timings reflect the end-to-end fit() call in MKLpy, which includes not only kernel averaging but also the downstream SVM training. In some datasets, the simple average may yield a kernel that is less aligned with the task, and the SVM then converges more slowly. We have been in correspondence with the authors of the MKLpy package and they have confirmed this explanation.. We added a paragraph in the Q4 section explaining this.
>
>
> Requested change 5:
>
> Thank you for this comment. We certainly agree that this is a strong assumption that in general will not hold in practice. It is for this reason that in the original manuscript, we started the paragraph immediately preceding the statement of Theorem 3 by explicitly stating that the simultaneously diagonalizeable setting is a special case. In the revised manuscript, we have added additional prose following the presentation of Theorem 3 to further highlight that the assumption is unlikely to hold in practice.
>
>
> Requested change 6:
>
> Thank you for this comment. We have address this in our revised manuscript.
>
>
> Requested change 7:
>
> Thank you for this comment. Note however that “a useful” is the correct grammatical structure while “an useful” is not correct. See for instance these references: https://www.dailywritingtips.com/a-useful-reminder-about-an/, https://dictionary.cambridge.org/us/dictionary/english/useful.
>
>
> Requested change 8:
>
> Thank you for this comment. We have address this in our revised manuscript.
>
>
> Requested change 9:
>
> Thank you for this comment. We have address this in our revised manuscript.
>
>
> Requested change 10:
>
> Thank you for this comment. We have address this in our revised manuscript.

---

### Comment · Action_Editor_n9KX · 2025-10-25
**Discussion**

Dear Reviewers,

Thank you very much for writing the detailed reviews. The authors have posted their responses to your reviews and submitted a revision. Would you please take a look and see whether you are satisfied with their responses and revisions? Thank you very much in advance.

Best regards,
AC

---

### Decision · Action_Editor_n9KX · 2025-11-16

**Recommendation:** Accept as is

**Audience:**

Yes

**Audience Explanation:**

Kernel learning is an efficient method to learn a good feature combination to improve the prediction ability. The paper proposes a new method to sparse combinations of kernels, which can improve the interpretability. The authors also use various optimization techniques in the algorithm design, such as convex optimization, dual problems and integer programming. These techniques should be interesting to the machine learning and optimization community.

**Claims And Evidence:**

Yes

**Claims Explanation:**

The paper proposes sparse kernel learning algorithms, and proposes to update the kernel model and kernel combination coefficients by alternating optimization. These updating strategies are supported by mathematical analysis and existing methodologies. For example, the paper shows that the kernel combination coefficient can be efficiently updated, and reduces the kernel model update to existing problems where one can use existing solvers. The paper also includes extensive experimental comparisons on various datasets. The results justify the effectiveness of the proposed method. All the reviewers think the paper is of good quality, and are satisfied with the authors' responses.